# Front-door Adjustment Beyond Markov Equivalence with Limited Graph Knowledge

**Abhin Shah**
Massachusetts Institute of Technology
abhin@mit.edu

**Karthikeyan Shanmugam**
Google Research
karthikeyanvs@google.com

**Murat Kocaoglu**
Purdue University
mkocaoglu@purdue.edu

## Abstract

Causal effect estimation from data typically requires assumptions about the cause-effect relations either explicitly in the form of a causal graph structure within the Pearlian framework, or implicitly in terms of (conditional) independence statements between counterfactual variables within the potential outcomes framework. When the treatment variable and the outcome variable are confounded, front-door adjustment is an important special case where, given the graph, causal effect of the treatment on the target can be estimated using *post-treatment* variables. However, the exact formula for front-door adjustment depends on the structure of the graph, which is difficult to learn in practice. In this work, we provide testable conditional independence statements to compute the causal effect using front-door-like adjustment without knowing the graph under limited structural side information. We show that our method is applicable in scenarios where knowing the Markov equivalence class is not sufficient for causal effect estimation. We demonstrate the effectiveness of our method on a class of random graphs as well as real causal fairness benchmarks.

## 1 Introduction

Causal effect estimation is at the center of numerous scientific, societal, and medical questions [Nabi et al., 2019, Castro et al., 2020]. The $do(\cdot)$ operator of Pearl represents the effect of an experiment on a causal system. For example, the probability distribution of a target variable $y$ after setting a treatment $t$ to $t$ is represented by $\mathbb{P}(y|do(t = t))$ and is known as an interventional distribution. Learning this distribution for any realization $t = t$[1] is what causal effect estimation entails. This distribution is different from the conditional distribution $\mathbb{P}(y|t = t)$ as there may be unobserved confounders between treatment and outcome that cannot be controlled for.

A causal graph, often depicted as a directed acyclic graph, captures the cause-and-effect relationships between variables and explains the causal system under consideration. A semi-Markovian causal model represents a causal model that includes unobserved variables influencing multiple observed variables [Verma and Pearl, 1990, Acharya et al., 2018]. In a semi-Markovian graph, directed edges between observed variables represent causal relationships, while bi-directed edges between observed variables represent unobserved common confounding (see Figure 1). Given any semi-Markovian

---

[1]Depending on the context, causal effect estimation sometimes refers to the estimating the difference of assigning $t = 1$ vs. $t = 0$ on the target variable $y$, e.g., $\mathbb{E}[y|do(t = 1)] - \mathbb{E}[y|do(t = 0)]$. This quantity is computable if we can identify $\mathbb{P}(y|do(t = t))$ for $t = \{0, 1\}$.

37th Conference on Neural Information Processing Systems (NeurIPS 2023).

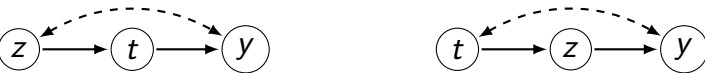

Figure 1: Representative graphs for back-door adjustment (left) and front-door adjustment (right).

graph, complete identification algorithms for causal effect estimation are known. For example, if $\mathbb{P}(y|do(t = t))$ is uniquely determined by the observational distribution and the causal graph, the algorithm by Shpitser and Pearl [2006] utilizes the graph to derive an *estimand*, i.e., the functional form mapping the observational distribution to the interventional distribution.

Certain special cases of estimands have found widespread use across several domains. One such special case is the *back-door adjustment* [Pearl, 1993] shown in Figure 1(left). The back-door adjustment utilizes the pre-treatment variable **z** (that blocks back-door paths) to control for unobserved confounder as follows:

$$\mathbb{P}(y|do(t = t)) = \sum_z \mathbb{P}(y|t = t, \mathbf{z} = \boldsymbol{z})\mathbb{P}(\mathbf{z} = \boldsymbol{z}), \tag{1}$$

where the do-calculus rules of Pearl [1995] are used to convert interventional distributions into observational distributions by leveraging the graph structure. However, the back-door adjustment is often inapplicable, e.g., in the presence of an unobserved confounder between $t$ and $y$. Surprisingly, in such scenarios, it is sometimes possible to find the causal effect using the *front-door adjustment* [Pearl, 1995] shown in Figure 1(right). Utilizing the front-door variable **z**, the front-door adjustment estimates the causal effect from observational distributions using the following formula (which is also obtained through the do-calculus rules and the graph structure):

$$\mathbb{P}(y|do(t = t)) = \sum_z \Big( \sum_{t'} \mathbb{P}(y|t = t', \mathbf{z} = \boldsymbol{z})\mathbb{P}(t = t') \Big) \mathbb{P}(\mathbf{z} = \boldsymbol{z}|t = t) \tag{2}$$

Recently, front-door adjustment has gained popularity in analyzing real-world data [Glynn and Kashin, 2017, Bellemare et al., 2019, Hünermund and Bareinboim, 2019] due to its ability to utilize post-treatment variables to estimate effects even in the presence of confounding between $t$ and $y$. However, in general, front-door adjustment also relies on knowing the causal graph, which may not always be feasible, especially in domains with many variables.

An alternative approach uses observational data to infer a Markov equivalence class, which is a collection of causal graphs that encode the same conditional independence relations [Spirtes et al., 2000]. A line of work [Perkovic et al., 2018, Jaber et al., 2019] provide identification algorithms for causal effect estimation from partial ancestral graphs (PAGs) [Zhang, 2008], a prominent representation of the Markov equivalence class, whenever every causal graph in the collection shares the same causal effect estimand. However, learning PAGs from data is challenging in practice due to the sequential nature of their learning algorithms, which can propagate errors between tests [Strobl et al., 2019a]. Further, to the best of our knowledge, there is no existing algorithm that can incorporate side information, such as known post-treatment variables, into PAG structure learning.

In this work, we ask the following question: *Can the causal effect be estimated with a testable criteria on observational data by utilizing some structural side information without knowing the graph?*

Recent research has developed such testable criteria to enable back-door adjustment without knowing the full causal graph [Entner et al., 2013, Cheng et al., 2020, Gultchin et al., 2020, Shah et al., 2022]. These approaches leverage structural side information, such as a known and observed parent of the treatment variable $t$. However, no such results have been established for enabling front-door adjustment. We address this gap by focusing on the case of unobserved confounding between $t$ and $y$, where back-door adjustment is inapplicable. Traditionally, this scenario has been addressed by leveraging the presence of an instrumental variable [Mogstad and Torgovitsky, 2018] or performing sensitivity analysis [Veitch and Zaveri, 2020], both of which provide only bounds in the non-parametric case. In contrast, we achieve identifiability by utilizing structural side information.

**Contributions.** We propose a method for estimating causal effects without requiring the knowledge of causal graph in the presence of unobserved confounding between treatment and outcome. Our approach utilizes front-door-like adjustments based on post-treatment variables and relies on conditional independence statements that can be directly tested from observational data. We

require one structural side information which can be obtained from an expert and is less demanding than specifying the entire causal graph. We illustrate that our framework provides identifiability in random ensembles where existing PAG-based methods are not applicable. Further, we illustrate the practical application of our approach to causal fairness analysis by estimating the total effect of a sensitive attribute on an outcome variable using the German credit data with fewer structural assumptions. The source code of our implementation is available at `https://github.com/abhin-shah/FD-adjustment-with-limited-graph`.

## 1.1 Related Work

**Effect estimation from causal graphs/Markov equivalence Class:** The problem of estimating interventional distributions with the knowledge of the semi-Markovian model has been studied extensively in the literature, with important contributions such as Tian and Pearl [2002] and Shpitser and Pearl [2006]. Perkovic et al. [2018] presented a complete and sound algorithm for identifying valid adjustments from PAGs. Going beyond valid adjustments, Jaber et al. [2019] proposed a complete and sound algorithm for identifying causal effect from PAGs. However, our method can recover the causal effect in scenarios where these algorithms are inapplicable.

**Effect estimation via front-door adjustment with causal graph:** Several recent works have contributed to a better understanding of the statistical properties of front-door estimation [Kuroki, 2000, Kuroki and Cai, 2012, Glynn and Kashin, 2018, Gupta et al., 2021], proposed robust generalizations [Hünermund and Bareinboim, 2019, Fulcher et al., 2020], and developed procedures to enumerate all possible front-door adjustment sets [Jeong et al., 2022, Wienöbst et al., 2022]. However, all of these require knowing the underlying causal graph. By contrast, Bhattacharya and Nabi [2022] verified the front-door criterion without knowing the causal graph using Verma constraint-based methodology. While their method was limited to a small set of graphs, ours leverages conditional independence, making it applicable to a much broader class of graphs. We note that they're applicable in different settings, depending on what the analyst knows about the problem.

## 2 Preliminaries and Problem Formulation

**Notations.** For a sequence of realizations $r_1, \cdots, r_n$, we define $\boldsymbol{r} \triangleq \{r_1, \cdots, r_n\}$. For a sequence of random variables $r_1, \cdots, r_n$, we define $\mathbf{r} \triangleq \{r_1, \cdots, r_n\}$. Let $\mathbb{1}$ denote the indicator function.

**Semi-Markovian Model and Effect Estimation.** We consider a causal effect estimation task where $\mathbf{x}$ represents the set of observed features, $t$ represents the observed treatment variable, and $y$ represents the observed outcome variable. We denote the set of all observed variables jointly by $\mathcal{V} \triangleq \{\mathbf{x}, t, y\}$. Let $\mathcal{U}$ denote the set of unobserved features that could be correlated with the observed variables.

We assume $\mathcal{W} \triangleq \mathcal{V} \cup \mathcal{U}$ follows a semi-Markovian causal model [Tian and Pearl, 2002] as below.

**Definition 1.** *A semi-Markovian causal model (SMCM) $\mathcal{M}$ is specified as follows:*

1. *$\mathcal{G}$ is a directed acyclic graph (DAG) over the set of vertices $\mathcal{W}$ such that each element of the set $\mathcal{U}$ has no parents.*
2. *$\forall v \in \mathcal{V}$, let $\pi^{(o)}(v) \subseteq \mathcal{V}$ and $\pi^{(u)}(v) \subseteq \mathcal{U}$ denote the set of parent of $v$ in $\mathcal{V}$ and $\mathcal{U}$, respectively.*
3. *$\mathbb{P}(\mathbf{u})$ is the unobserved joint distribution over the unobserved features.*
4. *The observational distribution is given by $\mathbb{P}(\mathbf{v}) = \mathbb{E}_{\boldsymbol{u}} \Big[ \prod_{v \in \mathcal{V}} \mathbb{P}(v | \pi^{(o)}(v), \pi^{(u)}(v)) \Big]$.*
5. *The interventional distribution when the variables $\mathbf{r} \subset \mathcal{V}$ are set to a fixed value $\boldsymbol{r}$ is given by*

$$\mathbb{P}(\mathbf{v} | do(\mathbf{r} = \boldsymbol{r})) = \mathbb{1}_{\mathbf{r} = \boldsymbol{r}} \cdot \mathbb{E}_{\boldsymbol{u}} \Big[ \prod_{v \in \mathcal{V} \setminus \mathbf{r}} \mathbb{P}(v | \pi^{(o)}(v), \pi^{(u)}(v)) \Big]. \tag{3}$$

6. *For any $v_1, v_2 \in \mathcal{V}$, if $\pi^{(u)}(v_1) \cap \pi^{(u)}(v_2) \neq \emptyset$, then $v_1$ and $v_2$ have a bi-directed edge in $\mathcal{G}$.*

In this work, we are interested in the causal effect of $t$ on $y$, i.e., $\mathbb{P}(y | do(t = t))$. We define this formally by marginalizing all variables except $y$ in the interventional distribution in (3).

**Definition 2.** *The causal effect of $t$ (when forced to a value $t$) on $y$ is given by:*

$$\mathbb{P}(y | do(t = t)) = \sum_{\mathbf{v} \setminus \{y\}} \mathbb{P}\big(\mathbf{v} \setminus \{y\}, y | do(t = t)\big). \tag{4}$$

Next, we define the notion of average treatment effect for a binary treatment $t$.

**Definition 3.** *The average treatment effect (ATE) of a binary treatment $t$ on outcome $y$ is given by $ATE = \mathbb{E}[y|do(t=1)] - \mathbb{E}[y|do(t=0)]$.*

Next, we define when the causal effect (Definition 2) is said to be identifiable from the observational distribution and the causal graph.

**Definition 4.** *(Causal effect identifiability) Given an observational distribution $\mathbb{P}(\mathbf{v})$ and a causal graph $\mathcal{G}$, the causal effect $\mathbb{P}(y|do(t=t))$ is identifiable if it is identical for every semi-Markovian Causal model with $(a)$ same graph $\mathcal{G}$ and $(b)$ same observational distribution $\mathbb{P}(\mathbf{v})$.*

In a causal graph $\mathcal{G}$, a path is an ordered sequence of distinct nodes where each node is connected to the next in the sequence by an edge. A path starting at node $w_1$ and ending at node $w_2$ in $\mathcal{G}$ is *blocked* by a set $\mathbf{w} \subset \mathcal{W} \setminus \{w_1, w_2\}$ if there exists $w \in \mathbf{w}$ such that (a) $w$ is not a collider or (b) $w$ is a collider and neither $w$ nor any of it's descendant is in $\mathbf{w}$. Further, $w_1$ and $w_2$ are said to be *d-separated* by $\mathbf{w}$ in $\mathcal{G}$ if $\mathbf{w}$ blocks every path between $w_1$ and $w_2$ in $\mathcal{G}$. Let $w_1 \perp\!\!\!\perp_d w_2 | \mathbf{w}$ denote that $w_1$ and $w_2$ are d-separated by $\mathbf{w}$ in $\mathcal{G}$. Similarly, let $w_1 \perp_p w_2 | \mathbf{w}$ denote that $w_1$ and $w_2$ are conditionally independent given $\mathbf{w}$. We assume causal faithfulness, i.e., any conditional independence $w_1 \perp_p w_2 | \mathbf{w}$ implies a d-separation relation $w_1 \perp\!\!\!\perp_d w_2 | \mathbf{w}$ in the causal graph $\mathcal{G}$.

## 2.1 Adjustment using pre-treatment variables

It is common in causal effect estimation to consider pre-treatment variables, i.e., variables that occur before the treatment in the causal ordering, and identify sets of variables that are *valid adjustments*. Specifically, a set $\mathbf{z} \subset \mathcal{V}$ forms a valid adjustment if the causal effect can be written as $\mathbb{P}(y|do(t=t)) = \sum_{\mathbf{z}} \mathbb{P}(y|t=t, \mathbf{z}=z)\mathbb{P}(\mathbf{z}=z)$. In other words, a valid adjustment $\mathbf{z}$ averages an estimate of $y$ regressed on $t$ and $\mathbf{z}$ with respect to the marginal distribution of $\mathbf{z}$, A popular criterion to find valid adjustments is to find a set $\mathbf{z} \subset \mathcal{V}$ that satisfies the *back-door criterion* [Pearl, 2009]. Formally, a set $\mathbf{z}$ satisfies the back-door criterion if (a) it blocks all back-door paths, i.e., paths between $t$ and $y$ that have an arrow pointing at $t$ and (b) no element of $\mathbf{z}$ is a descendant of $t$. While, in general, back-door sets can be found with the knowledge of the causal graph, recent works (see the survey Cheng et al. [2022]) have proposed testable criteria for identifying back-door sets with some causal side information, without requiring the entire graph.

## 2.2 Adjustment using post-treatment variables

While back-door adjustment is widely used, there are scenarios where no back-door set exists, e.g., when there is an unobserved confounder between $t$ and $y$. If no back-door set can be found from the pre-treatment variables, Pearlian theory can be used to identify post-treatment variables, i.e., the variables that occur after the treatment in the causal ordering, to obtain a *front-door adjustment*.

**Definition 5** (*Front-door criterion*). *A set $\mathbf{z} \subset \mathcal{V}$ satisfies the front-door criterion with respect to $t$ and $y$ if (a) $\mathbf{z}$ intercepts all directed paths from $t$ to $y$ (b) all back-door paths between $t$ and $\mathbf{z}$ are blocked, and (c) all back-door paths between $\mathbf{z}$ and $y$ are blocked by $t$.*

If a set $\mathbf{z}$ satisfies the front-door criterion, then the causal effect can be written as

$$\mathbb{P}(y|do(t=t)) = \sum_{\mathbf{z}} \Big( \sum_{t'} \mathbb{P}(y|t=t', \mathbf{z}=z)\mathbb{P}(t=t') \Big) \mathbb{P}(\mathbf{z}=z|t=t). \tag{5}$$

Intuitively, front-door adjustment estimates the causal effect of $t$ on $y$ as a composition of two effects: $(a)$ the effect of $t$ on $\mathbf{z}$ and $(b)$ the effect of $\mathbf{z}$ on $y$. However, one still needs the knowledge of the causal graph $\mathcal{G}$ to find a set satisfying the front-door criterion.

Inspired by the progress in finding back-door sets without knowing the entire causal graph, we ask: *Can testable conditions be derived to identify front-door-like sets using only partial structural information about post-treatment variables?* To that end, we consider the following side information.

**Assumption 1.** *The outcome $y$ is a descendant of the treatment $t$.*

**Assumption 2.** *There is an unobserved confounder between the outcome $y$ and the treatment $t$.*

**Assumption 3.** $\mathbf{b}$, *the set of all children of the treatment $t$, is observed and known.*

Assumption 1 is a fundamental assumption in most causal inference works, as it forms the basis for estimating non-trivial causal effects. Without it, the causal effect would be zero. Assumption 2 rules out the existence of sets that satisfy the back-door criteria, necessitating a different way of estimating the causal effect. Assumption 3 captures our side information by requiring every children of the treatment to be known and observed. To contrast, the side information in data-driven works on back-door adjustment requires a parent of the treatment to be known and observed [Shah et al., 2022].

Our assumptions imply that $\mathbf{b}$ intercepts all the directed paths from $t$ to $y$. Given this, it is natural to ask whether $\mathbf{b}$ satisfies the front-door criterion (Definition 5). We note that, in general, this is not true. We illustrate this via Figure 2 where we provide a causal graph $\mathcal{G}^{toy}$ satisfying our assumptions. However, $\mathbf{b}$ is not a valid front-door set in $\mathcal{G}^{toy}$ as the back-door path between $\mathbf{b}$ and $y$ via $\mathbf{z}^{(i)}$ is not blocked by $t$. Therefore, estimating the causal effect by assuming $\mathbf{b}$ is a front-door set might not always give an unbiased estimate. In the next section, we leverage the given side information and provide testable conditions to identify front-door-like sets.

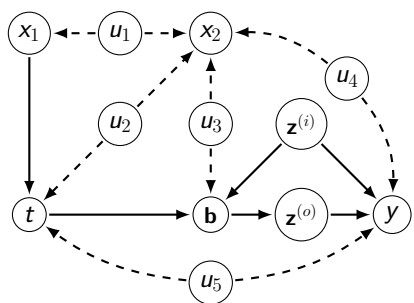

Figure 2: The graph $\mathcal{G}^{toy}$ satisfying Assumptions 1 to 3 where $u_i$ are unobserved.

# 3 Front-door Adjustment Beyond Markov Equivalence

In this section, we provide our main results, an algorithm for ATE estimation, and discuss the relationship to PAG-based methods. Our main results use observational criteria for causal effect estimation under Assumptions 1 to 3 using post treatment variables.

## 3.1 Causal effect estimation using post-treatment variables

First, we state a conditional independence statement implying causal identifiability. Then, we provide additional conditional independence statements resulting in a unique formula for effect estimation.

Causal identifiability (Definition 4) implies that the causal effect is uniquely determined given an observational distribution $\mathbb{P}(\mathcal{V})$ and the corresponding causal graph $\mathcal{G}$. We now show that satisfying a conditional independence statement (which can be tested solely from observational data, without requiring the graph $\mathcal{G}$) guarantees identifiability. We provide a proof in Appendix D.

**Theorem 3.1 (Causal Identifiability).** *Suppose Assumptions 1 to 3 hold. If there exists a set $\mathbf{z} \subseteq \mathcal{V} \setminus \{t, \mathbf{b}, y\}$ such that $\mathbf{b} \perp\!\!\!\perp_d y | t, \mathbf{z}$, then the causal effect of $t$ on $y$ is identifiable from observational data without the knowledge of the underlying causal graph $\mathcal{G}$.*

While the above result leads to identifiability, it does not provide a formula to compute the causal effect. In fact, the conditional independence $\mathbf{b} \perp\!\!\!\perp_d y | t, \mathbf{z}$ alone is *insufficient* to establish a unique formula, and different causal graphs lead to different formula. To illustrate this, we provide two SMCMs where Assumptions 1 to 3 and $\mathbf{b} \perp\!\!\!\perp_d y | t, \mathbf{z}$ hold, i.e., causal effect is identifiable from observational data via Theorem 3.1, but the formula is different. First, consider the SMCM in Figure 3(top) with $\mathbf{z} = (z_1, z_2)$ where causal effect is given by following formula (derived in Appendix D):

$$\mathbb{P}(y|do(t = t)) = \sum_{z_1, z_2, \mathbf{b}} \left( \sum_{t'} \mathbb{P}(y|z_1, z_2, t')\mathbb{P}(t'|z_1) \right) \times \mathbb{P}(z_2|\mathbf{b})\mathbb{P}(\mathbf{b}|t, z_1)\mathbb{P}(z_1). \qquad (6)$$

Next, consider the SMCM in Figure 3(bottom) with $\mathbf{z} = z_2$ where the causal effect is given by the front-door adjustment formula in (5) as $\mathbf{z}$ satisfies the front-door criterion. It remains to explicitly show that the formula in (6) is different from (5). To this end, we create a synthetic structural equation model (SEM) respecting the graph in Figure 3(top) and show that the formula in (5) gives a non-zero ATE error. In our SEM, the unobserved variable has a uniform distribution over $[1, 2]$. Each observed variable except $t$ is a sum of $(i)$ a linear combination of its parents with

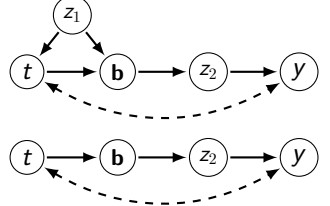

Figure 3: SMCMs on (top) & (bottom) satisfy $\mathbf{b} \perp\!\!\!\perp_d y | t, \mathbf{z}$ but have different causal effect estimation formulae.

coefficients drawn from uniform distribution over $[1, 2]$ and $(ii)$ a zero-mean Gaussian noise. The treatment variable is binarized by applying a standard logistic model to a linear combination of its parents with coefficients drawn as before. The ATE error averaged over 50 runs with 50000 samples in each run is $0.3842 \pm 0.0207$. See more experimental details in Appendix G.

Next, we provide two additional conditional independence statements that imply a unique formula for causal effect estimation. Our result is a *generalized front-door* with a formula identical to (5) as if $\mathbf{z}$ were a traditional front-door set. We also offer an alternative formula by utilizing a specific partition of $\mathbf{z}$ obtained from the conditional independence statements. We provide a proof in Appendix E.

**Theorem 3.2** (**A generalized front-door condition**). *Suppose Assumptions 1 to 3 hold. Let $\mathbf{z} \subseteq \mathcal{V} \setminus \{t, \mathbf{b}, y\}$ be a set satisfying*

$$\mathbf{b} \perp\!\!\!\perp_d y | t, \mathbf{z}, \tag{7}$$

*such that $\mathbf{z}$ can be decomposed into $\mathbf{z}^{(o)} \subseteq \mathbf{z}$ and $\mathbf{z}^{(i)} = \mathbf{z} \setminus \mathbf{z}^{(o)}$ with*

$$(i)\ \mathbf{z}^{(i)} \perp\!\!\!\perp_d t \quad and \quad (ii)\ \mathbf{z}^{(o)} \perp\!\!\!\perp_d t | \mathbf{b}, \mathbf{z}^{(i)}. \tag{8}$$

*Then, $\mathbf{z}$ and $\mathbf{s} \triangleq (\mathbf{b}, \mathbf{z}^{(i)})$ are generalized front-doors, and the causal effect of $t$ on $y$ can be obtained using any of the following equivalent formulae:*

$$\mathbb{P}(y|do(t = t)) = \sum_{\mathbf{z}} \Big( \sum_{t'} \mathbb{P}(y|\mathbf{z}, t')\mathbb{P}(t') \Big) \mathbb{P}(\mathbf{z}|t). \tag{9}$$

$$\mathbb{P}(y|do(t = t)) = \sum_{\mathbf{s}} \Big( \sum_{t'} \mathbb{P}(y|\mathbf{s}, t')\mathbb{P}(t') \Big) \mathbb{P}(\mathbf{s}|t). \tag{10}$$

**Remark 1.** *Consider the case where $y$ is a child of $t$. Then, the sufficient conditions in Theorems 3.1 and 3.2 do not hold because the conditional independence in (7) will not pass as $y \in \mathbf{b}$ (which follows from Assumption 3). However, in this case, the causal query itself is not identifiable as there exists a bi-directed edge between $y$ and $t$ (from Assumption 2) (see Tian and Pearl [2002, Theorem 4]).*

---

**Algorithm 1:** ATE estimation using subset search.

**Input:** $n_r, t, y, \mathbf{b}, \mathcal{Z}, p_v$
**Output:** $\text{ATE}_z, \text{ATE}_s$
**Initialization:** $\text{ATE}_z = 0, \text{ATE}_s = 0, c_1 = 0$
**for** $r = 1, \cdots, n_r$ **do** // Use a different train-test split in each run
    $\text{ATE}_z^r = 0, \text{ATE}_s^r = 0, c_2 = 0$; // These are used to average over different subsets that satisfy our conditions for a specific train-test split
    **for** $\mathbf{z} \in \mathcal{Z}$ **do** // Perform an exhaustive search over $\mathcal{Z}$
        **if** $CI(\mathbf{b} \perp_p y|\mathbf{z}, t) > p_v$ **then** // Check for (7) where $CI$ stands for conditional independence
            **for** $\mathbf{z}^{(o)} \subseteq \mathbf{z}$ **do** // Perform an exhaustive search over $\mathbf{z}$ to find $\mathbf{z}^{(o)}$ and $\mathbf{z}^{(i)}$
                $\mathbf{z}^{(i)} = \mathbf{z} \setminus \mathbf{z}^{(o)}$;
                **if** $\min\{CI(\mathbf{z}^{(i)} \perp_p t), CI(\mathbf{z}^{(o)} \perp_p t|\mathbf{b}, \mathbf{z}^{(i)})\} > p_v)$ **then** // Check for (8)
                    $c_2 = c_2 + 1, \mathbf{s} = (\mathbf{b}, \mathbf{z}^{(i)})$;
                    $\text{ATE}_z^r = \text{ATE}_z^r + \frac{\sum_{j:t_j=1} \sum_{t'} \mathbb{E}[y|\mathbf{z}_j, t']\mathbb{P}(t')}{|\{j:t_j=1\}|} - \frac{\sum_{j:t_j=0} \sum_{t'} \mathbb{E}[y|\mathbf{z}_j, t']\mathbb{P}(t')}{|\{j:t_j=0\}|}$;
                    $\text{ATE}_s^r = \text{ATE}_s^r + \frac{\sum_{j:t_j=1} \sum_{t'} \mathbb{E}[y|\mathbf{s}_j, t']\mathbb{P}(t')}{|\{j:t_j=1\}|} - \frac{\sum_{j:t_j=0} \sum_{t'} \mathbb{E}[y|\mathbf{s}_j, t']\mathbb{P}(t')}{|\{j:t_j=0\}|}$;
    **if** $c_2 > 0$ **then**
        $\text{ATE}_z = \text{ATE}_z + \text{ATE}_z^r/c_2, \text{ATE}_s = \text{ATE}_s + \text{ATE}_s^r/c_2, c_1 = c_1 + 1$;
**if** $c_1 > 0$ **then**
    $\text{ATE}_z = \text{ATE}_z/c_1, \text{ATE}_s = \text{ATE}_s/c_1$; // $c_1$ is used to average over different train-test splits.
**else**
    Failed to find $\mathbf{z} = (\mathbf{z}^{(i)}, \mathbf{z}^{(o)})$ satisfying (7) and (8);

## 3.2 Algorithm for ATE estimation

The ATE can be computed by taking the first moment version of (9) or (10). In Algorithm 1, we provide a systematic way to estimate the ATE using Theorem 3.2 by searching for a set $\mathbf{z} \in \mathcal{Z} \triangleq \mathcal{V} \setminus \{t, \mathbf{b}, y\}$ such that (a) p-value of conditional independence in (7) passes a threshold $p_v$ and (b) there exists a decomposition $\mathbf{z} = (\mathbf{z}^{(i)}, \mathbf{z}^{(o)})$ such that p-values of conditional independencies in (8) pass the threshold $p_v$. Then, for every such $\mathbf{z}$, the algorithm computes the ATE using the first moment version of (9), and averages. The algorithm produces another estimate by using (10) instead of (9).

## 3.3 Relation to PAG-based algorithms

Now, we exhibit how our approach can recover the causal effect in certain scenarios where PAG-based methods are not suitable. PAGs depict ancestral relationships (not necessarily direct) with directed edges and ambiguity in orientations (if they exist across members of the equivalence class) by circle marks. Figure 4(c) shows the PAG consistent with SMCM in Figure 4(a). While we formally define PAGs in Appendix B, we refer interested readers to Triantafillou and Tsamardinos [2015]. The IDP algorithm of Jaber et al. [2019] is sound and complete for identifying causal effect from PAGs.

Consider SMCM in Figure 4(a) where our approach recovers the causal effect as $(i)$ Assumptions 1 to (3), $(ii)$ (7), and $(iii)$ (8) hold (where $(ii)$ and $(iii)$ can be tested from observational data). However, the IDP algorithm fails to recover the effect from the PAG. To see this, consider SMCM in Figure 4(b) which is Markov equivalent to SMCM in Figure 4(a), i.e., the PAG in Figure 4(c) is also consistent with SMCM in Figure 4(b). Intuitively, when the strength of the edge between $t$ and $\mathbf{b}$ is very small but the strength of the edge between $t$ and $y$ is very high for both Figure 4(a) and Figure 4(b), causal effect in Figure 4(b) remains high while the causal effect in Figure 4(a) goes to zero.

We note that Assumptions 1 and 3, and (7) do not hold for the SMCM in Figure 4(b).

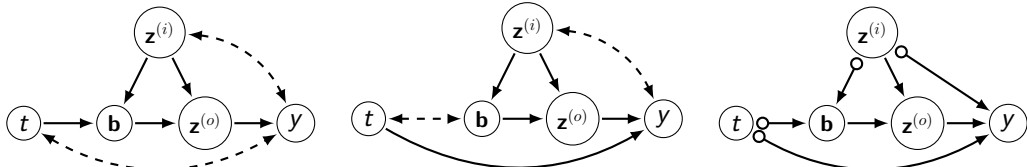

Figure 4: (a) An SMCM satisfying (7) and (8). (b) An SMCM obtained from (a) by modifying the edges between $t$ and $\mathbf{b}$ and between $t$ and $y$. (c) The PAG corresponding to SMCM in (a) and (b).

**Remark 2.** *Obtaining a PAG requires a large number of sequential conditional independence tests where the choice of the next test depends on the previous ones [Claassen et al., 2013]. The erroneous tests and orientation steps can potentially alter the structure of the PAG non-locally (see Strobl et al. [2016] for an example). This makes it difficult to control the false discovery rate for PAG based methods. Moreover, incorporating arbitrary side information into a PAG in a systematic way is still an open problem. In contrast, our approach does not rely on constructing a graphical object such as a PAG and the conditional independence tests could be carried out in parallel. Thus, our method can be viewed as a way to mitigate the issues associated with sequential testing by using structural side information. However, in scenarios where the structural side information is not available, we may have to resort to PAG-based methods.*

## 4 Empirical Evaluation

We evaluate our approach empirically in 3 ways: $(i)$ we demonstrate the applicability of our method on a class of random graphs, $(ii)$ we assess the effectiveness of our method in estimating the ATE using finite samples, and $(iii)$ we showcase the potential of our method for causal fairness analysis.

### 4.1 Applicability to a class of random graphs

In this experiment, we create a class of random SMCMs, sample 100 SMCMs from this class, and check if (7) and (8) hold by checking for corresponding d-separations in the SMCMs.

**Creation of random SMCMs.** Let $p \triangleq |\mathcal{V}|$ denote the dimension of observed variables including $\mathsf{x}$, $t$, and $y$. Let $v_1, \cdots, v_p$ denote a causal ordering of these variables. Our random ensemble depends on two parameters: $(i)$ $d \leq p/2$ which is the expected in-degree of variables $v_{2d}, \cdots, v_p$ and $(ii)$ $q \leq p$ which controls the number of unobserved features. For $1 \leq i < j \leq p$, we add $v_i \longrightarrow v_j$ with probability 0.5 if $j \leq 2d$ and with probability $d/(j-1)$ if $j > 2d$. We note that this procedure is such that the expected in-degree of variables $v_{2d}, \cdots, v_p$ is same (and equal to $d$) which is consistent with other recent work (e.g., Addanki et al. [2020]). Next, for $1 \leq i < j \leq p$, we add $v_i \leftrightarrow v_j$ with probability $q/p$ such that the expected number of unobserved features is $q(p-1)/2$. Then, we choose $v_p$ as $y$, any variable that is ancestor of $y$ but not its parent or grandparent as $t$, and all children of $t$ as $\mathbf{b}$. Finally, we add $t \leftrightarrow y$ if missing.

**Results.** We compare the success rate of two approaches: $(i)$ exhaustive search for $\mathbf{z}$ satisfying (7) and (8) which is exponential in $p$ and $(ii)$ search for a $\mathbf{z}$ of size at-most 5 satisfying (7) and (8) which is polynomial in $p$. We provide the number of successes of these approaches as a tuple in Table 1 for various $p$, $d$, and $q$. We see that the two approaches have comparable performances. We also compare with the IDP algorithm by providing it the true PAG. However, it gives 0 successes across various $p$, $d$, and $q$. We provide results for another random ensemble in Appendix G.

Table 1: Number of successes out of 100 random graphs for our methods shown as a tuple. The first method searches a $\mathbf{z}$ exhaustively and the second method searches a $\mathbf{z}$ with size at-most 5.

|  | $p = 10$ | | | $p = 15$ | | |
|  | $d = 2$ | $d = 3$ | $d = 4$ | $d = 2$ | $d = 3$ | $d = 4$ |
| --- | --- | --- | --- | --- | --- | --- |
| $q = 0.0$ | $(43, 43)$ | $(20, 20)$ | $(21, 21)$ | $(27, 26)$ | $(9, 9)$ | $(4, 2)$ |
| $q = 0.5$ | $(23, 23)$ | $(16, 16)$ | $(7, 7)$ | $(18, 17)$ | $(4, 3)$ | $(0, 0)$ |
| $q = 1.0$ | $(6, 6)$ | $(4, 4)$ | $(5, 5)$ | $(9, 9)$ | $(10, 9)$ | $(0, 0)$ |

## 4.2 Estimating the ATE

In this experiment, we generate synthetic data using the 6 random SMCMs in Section 4.1 for $p = 10$, $d = 2$, and $q = 1.0$ where our approach was successful indicating existence of $\mathbf{z} = (\mathbf{z}^{(i)}, \mathbf{z}^{(o)})$ such that the conditional independence statements in Theorem 3.2 hold. Then, we use Algorithm 1 to compute the error in estimating ATE and compare against a `Baseline` which uses the front-door adjustment in (5) with $\mathbf{z} = \mathbf{b}$ given the side information in Assumption 3. We provide the results for the same experiment for specific choices of SMCMs including the one in Figure 2 in Appendix G. We also provide the 6 random SMCMs in Appendix G. We use RCoT hypothesis test [Strobl et al., 2019b] for conditional independence testing from finite data.

**Data generation.** We use the following procedure to generate data from every SMCM. We generate unobserved variables independently from $\mathrm{Unif}[1, 2]$ which denotes the uniform distribution over $[1, 2]$. For every observed variable $v \in \mathcal{V}$, let $\pi(v) \triangleq (\pi^{(o)}(v), \pi^{(u)}(v)) \in \mathbb{R}^{d_v \times 1}$ denote the set of observed and unobserved parents of $v$ stacked as a column vector. Then, we generate $v \in \mathcal{V}$ as

$$v = \mathbf{a}_v^\top \pi(v) + 0.1 \ \mathcal{N}(0, 1) \text{ for } v \in \mathcal{V} \setminus \{t\} \quad \text{and} \quad t = \mathrm{Bernoulli}(\mathrm{Sigmoid}(\mathbf{a}_t^\top \pi(t))) \quad (11)$$

where the coefficients $\mathbf{a}_v \in \mathbb{R}^{d_v \times 1}$ with every entry sampled independently from $\mathrm{Unif}[1, 2]$. Also, to generate the true ATE, we intervene on the generation model in (11) by setting $t = 0$ and $t = 1$.

**Results.** For every SMCM, we generate $n$ samples of every observed variable in every run of the experiment. We average the ATE error over 10 such runs where the coefficients in (11) vary across runs. We report the average of these averages over the 6 SMCMs in Figure 5 for various $n$. While the error rates of `Baseline` and Algorithm 1 are of the similar order for $n = 100$, Algorithm 1 gives much lower errors for $n = 1000$ and $n = 10000$ showing the efficacy of our method.

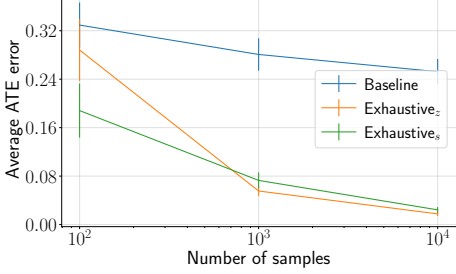

Figure 5: Average ATE for Algorithm 1 and `Baseline` vs. number of samples.

## 4.3 Experiments with real-world fairness benchmarks

Next, we describe how our results enable finding front-door-like adjustment sets in fairness problems. In a typical fairness problem, the goal is to ensure that the outcome variable $y$ does not unfairly depend on the sensitive/protected attribute, e.g., race or gender (which we define to be treatment variable $t$), which would reflect undesirable biases. Often, the outcome is a descendant of the sensitive attribute (as per Assumption 1), and both outcome and sensitive attribute are confounded by unobserved variables (as per Assumption 2). Furthermore, there are be a multitude of measured post-sensitive-attribute variables that can affect the outcome. This stands in contrast to the usual settings for causal effect estimation, where pre-treatment variables are primarily utilized.

Fairness problems are typically evaluated using various fairness metrics, such as causal fairness metrics or observational metrics. Causal metrics require knowing the underlying causal graph, which can be a challenge in practice. Observational criteria can be decomposed into three types of effects [Zhang and Bareinboim, 2018, Plecko and Bareinboim, 2022]: spurious effects, direct effects, and indirect effects (through descendants of sensitive attribute). In some scenarios, capturing the sum of direct and indirect effects is of interest, but even this requires knowing the causal graph.

Now, we demonstrate the application of our adjustment formulae in Theorem 3.2 to compute the sum of direct and indirect effects of the sensitive attribute on the outcome, while separating it from spurious effects. The sum of these effects is indeed the causal effect of sensitive attribute on the outcome. In other words, we consider the following fairness metric: $\mathbb{E}[y|do(t=1)] - \mathbb{E}[y|do(t=0)]$. We assume that all the children of the sensitive attribute are known, which may be easier to justify compared to the typical assumption in causal fairness literature of knowing the entire causal graph.

**German Credit Dataset.** The German Credit dataset [Hofmann, 1994] is used for credit risk analysis where the goal is to predict whether a loan applicant is a good or bad credit risk based on applicant's 20 demographic and socio-economic attributes. The binary credit risk is the outcome $y$ and the applicant's age (binarized by thresholding at 25 [Kamiran and Calders, 2009]) is the sensitive attribute $t$. Further, the categorical attributes are one-hot encoded.

We apply Algorithm 1 with $n_r = 100$ and $p_v = 0.1$ where we search for a set $\mathbf{z} = (\mathbf{z}^{(o)}, \mathbf{z}^{(i)})$ of size at most 3 under the following two distinct assumptions on the set of all children $\mathbf{b}$ of $t$:

1. When considering $\mathbf{b} = \{$# of people financially dependent on the applicant, applicant's savings, applicant's job$\}$, Algorithm 1 results in $\mathbf{z}^{(i)} = \{$purpose for which the credit was needed, indicator of whether the applicant was a foreign worker$\}$, $\mathbf{z}^{(o)} = \{$installment plans from providers other than the credit-giving bank$\}$, $\text{ATE}_z = 0.0125 \pm 0.0011$, and $\text{ATE}_s = 0.0105 \pm 0.0018$.

2. When considering $\mathbf{b} = \{$# of people financially dependent on the applicant, applicant's savings$\}$, Algorithm 1 results in $\mathbf{z}^{(i)} = \{$purpose for which the credit was needed, applicant's checking account status with the bank$\}$, $\mathbf{z}^{(o)} = \{$installment plans from providers other than the credit-giving bank$\}$, $\text{ATE}_z = 0.0084 \pm 0.0008$, and $\text{ATE}_s = -0.0046 \pm 0.0021$.

Under the first assumption above, the causal effect using the adjustment formulae in (9) and (10) have same sign and are close in magnitude. However, under the second assumption, the effect flips sign. The results suggest that the second hypothesis regarding $\mathbf{b}$ is incorrect, implying that applicant's job may indeed be a direct child of applicant's age, which aligns with intuition.

The dataset has only 1000 samples, which increases the possibility of detecting independencies in our criterion by chance, even with the size of $\mathbf{z}$ constrained. To address this issue, we use 100 random bootstraps with a sample size equal to half of the training data and evaluate the p-value of our conditional independence criteria for all subsets returned by our algorithm. We select the subset $\mathbf{z}$ with the highest median p-value (computed over the bootstraps) and use it in our adjustment formulae on a held out test set. To assess the conditional independencies associated with the selected $\mathbf{z}$, we plot a histogram of the corresponding p-values for all these bootstraps. If the conditional independencies hold, we expect the p-values to be spread out, which we observe in the histograms in Figure 6 for the first choice of $\mathbf{b}$. We report similar results for the second choice of $\mathbf{b}$ in Appendix G.

**Adult Dataset:** We perform a similar analysis on the Adult dataset [Kohavi and Becker, 1996]. With suitable choices of $\mathbf{b}$, Algorithm 1 was unable to find a suitable $\mathbf{z}$ satisfying $\mathbf{b} \perp_p y|\mathbf{z}, t$. This suggests that in this dataset, there may not be any non-child descendants of the sensitive attribute, which is required for our criterion to hold. More details can be found in Appendix G.

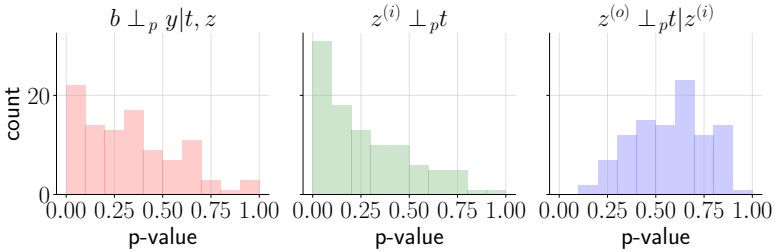

Figure 6: Histograms of p-values of the conditional independencies in (9) and (10) over 100 bootstrap runs for $\mathbf{b} = \{$# of people financially dependent on the applicant, applicant's savings, applicant's job$\}$.

## 5 Conclusion, Limitations, and Future Work

In this work, we proposed sufficient conditions for causal effect estimation through a generalized front-door adjustment given structural side information irrespective of the number or complexity of the latent confounders. Our approach can identify causal effect in graphs where known Markov equivalence classes do not allow identification. However, our approach relies primarily on two assumptions: Assumption 2 and Assumption 3.

Assumption 2 plays a crucial role in Theorem 3.2 (see the discussion in Section E.3) as it requires the presence of an unobserved confounder between the treatment variable and the outcome variable. This assumption is necessary for the applicability of our approach. If Assumption 2 does not hold, it implies that there is a set that satisfies the back-door criterion, and existing methods for finding back-door adjustment sets [Entner et al., 2013, Cheng et al., 2020, Shah et al., 2022] can be utilized. This suggests that our results could be derived under the weaker condition that there is an unblockable back-door path between $t$ and $y$. However, in many real-world scenarios, the presence of unobserved variables that potentially confound the treatment and the outcome is common, and we expect Assumption 2 to be true.

Assumption 3 is the requirement of knowing the entire set of children of the treatment variable within the causal graph. While this is strictly less demanding than specifying the entire causal graph, it may still present practical challenges in real-world scenarios. For instance, in large-scale observational studies or domains with numerous variables, exhaustively identifying all the children may be computationally demanding. It remains unclear whether one can estimate the causal effect using front-door-like adjustment with even less side information, e.g., knowing only one child of the treatment variable or knowing any subset of children of the treatment variable. An important future direction could be to approximate the causal effect when only the children corresponding to weak edges are unknown. Such variations around our condition are promising directions for future work. However, until then, one could seek input from domain experts. These experts possess valuable knowledge and insights about the specific domain under study, which can aid in identifying all the relevant variables that serve as children of the treatment.

The time complexity of Algorithm 1 is exponential due to its search over all possible subsets of observed variables (except $t, \mathbf{b}, y$). While this is inherent in the general case, recent work by Shah et al. [2022] proposed a scalable approximation for conditional independence testing using continuous optimization by exploiting the principle of invariant risk minimization, specifically for back-door adjustment without the need for the causal graph. However, extending this approach to multiple conditional independence tests, as required in (7) and (8), remains an open challenge. Therefore, exploring the development of continuous optimization-based methods for scalability of front-door adjustment in the absence of the causal graph is crucial. Further, Algorithm 1 could potentially be augmented with ideas from double machine learning and inverse variance weighting for bias and variance reduction.

Lastly, it is important to note that Theorem 3.2 provides only sufficient conditions for causal effect estimation. This means that there may be cases where front-door-like adjustment is possible, but the conditions stated in Theorem 3.2 do not hold. Specifically, there could be scenarios, such as when the outcome variable is a child of the children of the treatment variable, i.e., $y$ is a child of $\mathbf{b}$, where conditions (7) and (8) are not satisfied.

## Acknowledgements

We thank the anonymous reviewers for their comments and suggestions. Murat Kocaoglu acknowledges the support of NSF Grant CAREER 2239375.

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

## A   Broader impact

In this work, we propose an algorithm for estimating causal effects from observational studies without relying on expert knowledge of the causal model. Our approach is particularly valuable in scenarios where conducting randomized control trials (RCTs) is challenging or unethical, such as in healthcare settings where consensus treatment protocols are often determined based on observational data. While our algorithm shows promise in providing accurate causal effect estimates, it is crucial to address the potential negative impact of incorrect results that may arise from our work.

One significant concern is the possibility of miscalculating the treatment effect due to limitations in testing power at finite sample sizes or the misidentification of certain features as direct children of the treatment variable. This introduces the risk of inaccurate estimations, which could have detrimental consequences when making decisions or establishing treatment protocols based on the conclusions derived from our algorithm. It is essential to approach the interpretation of our algorithm's results with caution and subject them to critical evaluation. It is worth noting that the potential for incorrect results is not unique to our algorithm but is inherent in most observational studies and effect estimation algorithms. Acknowledging these potential negative impacts emphasizes the need for further research to improve the reliability and accuracy of causal effect estimation in observational studies.

## B   Preliminaries about ancestral graphs

In this section, we provide the definition of *partial ancestral graphs* (PAGs). PAGs are defined using *maximal ancestral graphs* (MAGs). Below, we define MAGs and PAGs based on their construction from directed acyclic graphs (DAGs).

A MAG can be obtained from a DAG as follows: if two observed nodes $x_1$ and $x_2$ cannot be d-separated conditioned on any subset of observed variables, then $(i)$ $x_1 \longrightarrow x_2$ is added in the MAG if $x_1$ is an ancestor of $x_2$ in the DAG, $(ii)$ $x_2 \longrightarrow x_1$ is added in the MAG if $x_2$ is an ancestor of $x_1$ in the DAG, and $(iii)$ $x_1 \leftrightarrow x_2$ is added in the MAG if $x_1$ and $x_2$ are not ancestrally related in the DAG. $(iv)$ After the above three operations, if both $x_1 \leftrightarrow x_2$ and $x_1 \longrightarrow x_2$ are present, we retain only the directed edge. In general, a MAG represents a collection of DAGs that share the same set of observed variables and exhibit the same independence and ancestral relations among these observed variables. It is possible for different MAGs to be Markov equivalent, meaning they represent the exact same independence model.

A PAG shares the same adjacencies as any MAG in the observational equivalence class of MAGs. An end of an edge in the PAG is marked with an arrow ($>$ or $<$) if the edge appears with the same arrow in all MAGs in the equivalence class. An end of an edge in the PAG is marked with a circle ($o$) if the edge appears as an arrow ($>$ or $<$) and a tail ($-$) in two different MAGs in the equivalence class.

## C   Rules of do-calculus

In this section, we provide the do-calculus rules of Pearl [1995] that are used to prove our main results in the following sections. We build upon the definition of semi-Markovian causal model from Section 2.

For any $\mathbf{v} \in \mathcal{W}$, let $\mathcal{G}_{\overline{\mathbf{v}}}$ be the graph obtained by removing the edges going into $\mathbf{v}$ in $\mathcal{G}$, and let $\mathcal{G}_{\underline{\mathbf{v}}}$ be the graph obtained by removing the edges going out of $\mathbf{v}$ in $\mathcal{G}$.

**Theorem C.1** (Rules of do-calculus, Pearl [1995]). *For any disjoint subsets* $\mathbf{v}_1, \mathbf{v}_2, \mathbf{v}_3, \mathbf{v}_4 \subseteq \mathcal{W}$, *we have the following rules.*

*Rule 1:* $\mathbb{P}(\mathbf{v}_1|do(\mathbf{v}_2), \mathbf{v}_3, \mathbf{v}_4) = \mathbb{P}(\mathbf{v}_1|do(\mathbf{v}_2), \mathbf{v}_3)$      *if* $\mathbf{v}_1 \perp\!\!\!\perp_d \mathbf{v}_4|\mathbf{v}_2, \mathbf{v}_3$ *in* $\mathcal{G}_{\overline{\mathbf{v}}_2}$.

*Rule 2:* $\mathbb{P}(\mathbf{v}_1|do(\mathbf{v}_2), \mathbf{v}_3, do(\mathbf{v}_4)) = \mathbb{P}(\mathbf{v}_1|do(\mathbf{v}_2), \mathbf{v}_3, \mathbf{v}_4)$      *if* $\mathbf{v}_1 \perp\!\!\!\perp_d \mathbf{v}_4|\mathbf{v}_2, \mathbf{v}_3$ *in* $\mathcal{G}_{\overline{\mathbf{v}}_2, \underline{\mathbf{v}}_4}$.

*Rule 3:* $\mathbb{P}(\mathbf{v}_1|do(\mathbf{v}_2), \mathbf{v}_3, do(\mathbf{v}_4)) = \mathbb{P}(\mathbf{v}_1|do(\mathbf{v}_2), \mathbf{v}_3)$      *if* $\mathbf{v}_1 \perp\!\!\!\perp_d \mathbf{v}_4|\mathbf{v}_2, \mathbf{v}_3$ *in* $\mathcal{G}_{\overline{\mathbf{v}}_2, \overline{\mathbf{v}_4(\mathbf{v}_3)}}$,

*where* $\mathbf{v}_4(\mathbf{v}_3)$ *is the set of nodes in* $\mathbf{v}_4$ *that are not ancestors of any node in* $\mathbf{v}_3$ *in* $\mathcal{G}_{\overline{\mathbf{v}}_2}$. *Pearl [1995] also gave an alternative criterion for Rule 3.*

*Rule 3a:* $\mathbb{P}(\mathbf{v}_1|do(\mathbf{v}_2), \mathbf{v}_3, do(\mathbf{v}_4)) = \mathbb{P}(\mathbf{v}_1|do(\mathbf{v}_2), \mathbf{v}_3)$      *if* $\mathbf{v}_1 \perp\!\!\!\perp_d F_{\mathbf{v}_4}|\mathbf{v}_2, \mathbf{v}_3$ *in* $\mathcal{G}_{\overline{\mathbf{v}}_2}^{\mathbf{v}_4}$,

where $\mathcal{G}^{\mathbf{v}_4}$ is the graph obtained from $\mathcal{G}$ after adding $(a)$ a node $F_{\mathbf{v}_4}$ and $(b)$ edges from $F_{\mathbf{v}_4}$ to every node in $\mathbf{v}_4$.

Also, throughout our proofs, we use the following fact.

**Fact 1.** *Consider any $\mathcal{G}'$ obtained by removing any edge(s) from $\mathcal{G}$. For any sets of variables $\mathbf{v}_1, \mathbf{v}_2, \mathbf{v}_3 \subseteq \mathcal{W}$, if $\mathbf{v}_1$ and $\mathbf{v}_2$ are d-separated by $\mathbf{v}_3$ in $\mathcal{G}$ than $\mathbf{v}_1$ and $\mathbf{v}_2$ are d-separated by $\mathbf{v}_3$ in $\mathcal{G}'$.*

## D  Causal Identifiability

In this section, we derive the causal effect for the SMCM in Figure 3(top), i.e., (6), as well as prove Theorem 3.1 one by one.

### D.1  Proof of (6)

First, using the law of total probability, we have

$$\mathbb{P}(y|do(t=t)) = \sum_{z_1, z_2} \mathbb{P}(y|do(t=t), z_1 = z_1, z_2 = z_2)\mathbb{P}(z_1 = z_1, z_2 = z_2|do(t=t)). \tag{12}$$

Now, we show that the two terms in RHS of (12) can be simplified as follows

$$\mathbb{P}(y|do(t=t), z_1 = z_1, z_2 = z_2) = \sum_{t'} \mathbb{P}(y|z_1 = z_1, z_2 = z_2, t = t')\mathbb{P}(t = t'|z_1 = z_1). \tag{13}$$

$$\mathbb{P}(z_1 = z_1, z_2 = z_2|do(t=t)) = \sum_{b} \mathbb{P}(z_2 = z_2|\mathbf{b} = b)\mathbb{P}(\mathbf{b} = b|t = t, z_1 = z_1)\mathbb{P}(z_1 = z_1), \tag{14}$$

Combining (12) to (14) results in (6).

**Proof of (13):**  We have

$$\mathbb{P}(y|do(t=t), z_1 = z_1, z_2 = z_2) \tag{15}$$

$$\overset{(a)}{=} \mathbb{P}(y = y|do(t=t), z_1 = z_1, z_2 = z_2, \mathbf{b} = b) \tag{16}$$

$$\overset{(b)}{=} \mathbb{P}(y = y|do(t=t), z_1 = z_1, z_2 = z_2, do(\mathbf{b} = b)) \tag{17}$$

$$\overset{(c)}{=} \mathbb{P}(y = y|z_1 = z_1, z_2 = z_2, do(\mathbf{b} = b)) \tag{18}$$

$$\overset{(d)}{=} \sum_{t'} \mathbb{P}(y = y|z_1 = z_1, z_2 = z_2, do(\mathbf{b} = b), t = t')\mathbb{P}(t = t'|z_1 = z_1, z_2 = z_2, do(\mathbf{b} = b)) \tag{19}$$

$$\overset{(e)}{=} \sum_{t'} \mathbb{P}(y = y|z_1 = z_1, z_2 = z_2, t = t')\mathbb{P}(t = t'|z_1 = z_1, z_2 = z_2, do(\mathbf{b} = b)) \tag{20}$$

$$\overset{(f)}{=} \sum_{t'} \mathbb{P}(y = y|z_1 = z_1, z_2 = z_2, t = t')\mathbb{P}(t = t'|z_1 = z_1, do(\mathbf{b} = b)) \tag{21}$$

$$\overset{(g)}{=} \sum_{t'} \mathbb{P}(y = y|z_1 = z_1, z_2 = z_2, t = t')\mathbb{P}(t = t'|z_1 = z_1), \tag{22}$$

where $(a)$ and $(f)$ follow from Rule 1, $(b)$ follows from Rule 2, $(c)$, $(e)$, and $(g)$ follow from Rule 3a, and $(d)$ follows from the law of total probability.

**Proof of (14):**  From the law of total probability, we have

$$\mathbb{P}(z_1 = z_1, z_2 = z_2|do(t=t)) \tag{23}$$

$$= \sum_{b} \mathbb{P}(z_1 = z_1, z_2 = z_2|do(t=t), \mathbf{b} = b)\mathbb{P}(\mathbf{b} = b|do(t=t)) \tag{24}$$

$$\overset{(a)}{=} \sum_{b} \mathbb{P}(z_2 = z_2|do(t=t), \mathbf{b} = b)\mathbb{P}(z_1 = z_1|do(t=t), \mathbf{b} = b, z_2 = z_2)\mathbb{P}(\mathbf{b} = b|do(t=t)) \tag{25}$$

$$\overset{(b)}{=} \sum_b \mathbb{P}(z_2\!=\!z_2|\mathbf{b}=\boldsymbol{b})\mathbb{P}(z_1\!=\!z_1|do(t=t),\mathbf{b}=\boldsymbol{b},z_2\!=\!z_2)\mathbb{P}(\mathbf{b}=\boldsymbol{b}|do(t=t)) \tag{26}$$

$$\overset{(c)}{=} \sum_b \mathbb{P}(z_2\!=\!z_2|\mathbf{b}=\boldsymbol{b})\mathbb{P}(z_1\!=\!z_1|do(t=t),\mathbf{b}=\boldsymbol{b})\mathbb{P}(\mathbf{b}=\boldsymbol{b}|do(t=t)) \tag{27}$$

$$\overset{(d)}{=} \sum_b \mathbb{P}(z_2\!=\!z_2|\mathbf{b}=\boldsymbol{b})\mathbb{P}(z_1\!=\!z_1,\mathbf{b}=\boldsymbol{b}|do(t=t)) \tag{28}$$

$$\overset{(e)}{=} \sum_b \mathbb{P}(z_2\!=\!z_2|\mathbf{b}=\boldsymbol{b})\mathbb{P}(z_1\!=\!z_1|do(t=t))\mathbb{P}(\mathbf{b}=\boldsymbol{b}|do(t=t),z_1\!=\!z_1) \tag{29}$$

$$\overset{(f)}{=} \sum_b \mathbb{P}(z_2\!=\!z_2|\mathbf{b}=\boldsymbol{b})\mathbb{P}(z_1\!=\!z_1)\mathbb{P}(\mathbf{b}=\boldsymbol{b}|do(t=t),z_1\!=\!z_1) \tag{30}$$

$$\overset{(g)}{=} \sum_b \mathbb{P}(z_2\!=\!z_2|\mathbf{b}=\boldsymbol{b})\mathbb{P}(z_1\!=\!z_1)\mathbb{P}(\mathbf{b}=\boldsymbol{b}|t=t,z_1\!=\!z_1) \tag{31}$$

where $(a)$, $(d)$, and $(e)$ follow from the definition of conditional probability, $(b)$ and $(f)$ follows from Rule 3a, $(c)$ follows from Rule 1, and $(g)$ follows from Rule 2.

## D.2   Proof of Theorem 3.1

Let $\mathrm{An}(y)$ denote the union of $y$ and the set of ancestors of $y$, and let $\mathcal{G}^{\mathrm{An}(y)}$ denote the subgraph of $\mathcal{G}$ composed only of nodes in $\mathrm{An}(y)$. First, we show that if $\mathbf{b} \perp\!\!\!\perp_d y|t,\mathbf{z}$ holds for some $\mathbf{z}$, then there is no bi-directed path between $t$ to $\mathbf{b}$ in $\mathcal{G}^{\mathrm{An}(y)}$.

**Lemma 1.** *Suppose Assumptions 1 to 3 hold. Suppose there exists a set $\mathbf{z} \subseteq \mathcal{V} \setminus \{t,\mathbf{b},y\}$ such that $\mathbf{b} \perp\!\!\!\perp_d y|t,\mathbf{z}$. Then, there is no bi-directed path between $t$ and $\mathbf{b}$ in $\mathcal{G}^{\mathrm{An}(y)}$.*

Given this claim, Theorem 3.1 follows from Tian and Pearl [2002, Theorem 4]. It remains to prove Lemma 1.

**Proof of Lemma 1.** We prove this result by contradiction. First, from Assumptions 1 and 3, $t \in \mathrm{An}(y)$ and $\mathbf{b}_0 \in \mathrm{An}(y)$ for some $\mathbf{b}_0 \subset \mathbf{b}$. Assume there exists a bi-directed path between $t$ and some $b \in \mathbf{b}_0$ in $\mathcal{G}^{\mathrm{An}(y)}$. Let $\mathcal{P}(t,b)$ denote the shortest of these paths. This path is of the form $t \leftrightarrow v_1 \leftrightarrow \cdots \leftrightarrow v_r \leftrightarrow b$ for some $r \geq 0$ where $v_q \in \mathcal{G}^{\mathrm{An}(y)}$ for every $q \in [r]$. We have the following two cases depending on the value of $r$.

(i) $r = 0$: In this case, consider the path $\mathcal{P}(y,b) \supset \mathcal{P}(t,b)$ in $\mathcal{G}$ of the form: $y \leftrightarrow t \leftrightarrow b$ in $\mathcal{G}$ (such a path exists because of Assumption 2). The path $\mathcal{P}(y,b)$ is unblocked when $t$ and $\mathbf{z}$ are conditioned on contradicting $\mathbf{b} \perp\!\!\!\perp_d y|t,\mathbf{z}$.

(ii) $r \geq 1$: In this case, consider the path $\mathcal{P}(y,b) \supset \mathcal{P}(t,b)$ in $\mathcal{G}$ of the form: $y \leftrightarrow t \leftrightarrow v_1 \leftrightarrow \cdots \leftrightarrow v_r \leftrightarrow b$ (such a path exists because of Assumption 2). We have the following two scenarios depending on whether the path $\mathcal{P}(y,b)$ is unblocked or blocked when $t$ and $\mathbf{z}$ are conditioned on. Suppose we condition on $t$ and $\mathbf{z}$.

  (a) The path $\mathcal{P}(y,b)$ is unblocked: In this case, by assumption, $\mathbf{b} \perp\!\!\!\perp_d y|t,\mathbf{z}$ is contradicted.

  (b) The path $\mathcal{P}(y,b)$ is blocked: We create a set $\mathbf{w}$ such that for any $w \in \mathbf{w}$ the following are true: $(a)$ $w = v_q$ for some $q \in [r]$, $(b)$ $w \notin \mathbf{z}$, $(b)$ there is no descendant path $\mathcal{P}(w,z)$ between $w$ and some $z \in \mathbf{z}$, and $(c)$ there is no descendant path $\mathcal{P}(w,t)$ between $w$ and $t$.

In this scenario, $\mathbf{w} \neq \emptyset$ because $\mathcal{P}(y,b)$ is blocked. Let $w_c \in \mathbf{w}$ be that node which is closest to $b$ in the path $\mathcal{P}(y,b)$. By the choice of $w_c$, the path $\mathcal{P}(w_c,b) \subset \mathcal{P}(y,b)$ is unblocked (when $t$ and $\mathbf{z}$ are conditioned on). Furthermore, by the definition of $\mathbf{w}$, $(a)$ $w_c \in \mathcal{G}^{\mathrm{An}(y)}$ (because $w_c = v_q$ for some $q \in [r]$) and $(b)$ there exists a descendant path $\mathcal{P}(w_c,y)$ between $w_c$ and $y$ such that $t \notin \mathcal{P}(w_c,y)$ as well as $z \notin \mathcal{P}(w_c,y)$ for every $z \in \mathbf{z}$. Therefore, the path $\mathcal{P}(w_c,y)$ is unblocked (when $t$ and $\mathbf{z}$ are conditioned on). Consider the path $\mathcal{P}'(y,b)$ obtained after concatenating $\mathcal{P}(w_c,y)$ and $\mathcal{P}(w_c,b)$ at $w_c$. This path is unblocked (when $t$ and $\mathbf{z}$ are conditioned on) because: $(a)$ $\mathcal{P}(w_c,b)$ is unblocked, $(b)$ $\mathcal{P}(w_c,y)$ is unblocked, and $(c)$ there is no collider at $w_c$ in this path (because $\mathcal{P}(w_c,y)$ is a descendant path to $y$). However, this contradicts $\mathbf{b} \perp\!\!\!\perp_d y|t,\mathbf{z}$.

# E    A generalized front-door condition

In this section, we prove Theorem 3.2. We begin by stating a few d-separation statements used in this proof. See Appendix F for a proof.

**Lemma 2.** *Suppose Assumptions 1 to 3 and d-separation criteria in Theorem 3.2, i.e., (7) and (8), hold. Then,*

*(a)* $y \perp\!\!\!\perp_d F_t | \mathbf{z}, \mathbf{b}$ *in* $\mathcal{G}_{\underline{\mathbf{b}}}^{t}$ *and* $y \perp\!\!\!\perp_d F_t | \mathbf{z}^{(i)}, \mathbf{b}$ *in* $\mathcal{G}_{\underline{\mathbf{b}}}^{t}$,

*(b)* $t \perp\!\!\!\perp_d \mathbf{b}$ *in* $\mathcal{G}_{\underline{t}}$,

*(c)* $t \perp\!\!\!\perp_d \mathbf{z}^{(i)} | \mathbf{b}$ *in* $\mathcal{G}_{\underline{t}}$,

*(d)* $t \perp\!\!\!\perp_d F_{\mathbf{b}} | \mathbf{z}^{(i)}$ *in* $\mathcal{G}^{\mathbf{b}}$, *and*

*(e)* $y \perp\!\!\!\perp_d \mathbf{b} | t, \mathbf{z}^{(i)}$ *in* $\mathcal{G}_{\underline{\mathbf{b}}}$.

Now, we proceed with the proof in two parts. In the first part, we prove (9), and in the second part, we prove (10).

## E.1    Proof of (9)

First, using the law of total probability, we have

$$\mathbb{P}(y = y | do(t = t)) = \sum_{\mathbf{z}} \mathbb{P}(y = y | do(t = t), \mathbf{z} = \mathbf{z}) \mathbb{P}(\mathbf{z} = \mathbf{z} | do(t = t)). \tag{32}$$

Now, we show that the two terms in RHS of (32) can be simplified as follows

$$\mathbb{P}(y = y | do(t = t), \mathbf{z} = \mathbf{z}) = \sum_{t'} \mathbb{P}(y = y | \mathbf{z} = \mathbf{z}, t = t') \mathbb{P}(t = t'). \tag{33}$$

$$\mathbb{P}(\mathbf{z} = \mathbf{z} | do(t = t)) = \mathbb{P}(\mathbf{z} = \mathbf{z} | t = t), \tag{34}$$

Combining (33) and (34) completes the proof of (9).

**Proof of (33):**    We have

$$\mathbb{P}(y = y | do(t = t), \mathbf{z} = \mathbf{z}) \stackrel{(a)}{=} \mathbb{P}(y = y | do(t = t), \mathbf{z} = \mathbf{z}, \mathbf{b} = \mathbf{b}) \tag{35}$$

$$\stackrel{(b)}{=} \mathbb{P}(y = y | do(t = t), \mathbf{z} = \mathbf{z}, do(\mathbf{b} = \mathbf{b})) \tag{36}$$

$$\stackrel{(c)}{=} \mathbb{P}(y = y | \mathbf{z} = \mathbf{z}, do(\mathbf{b} = \mathbf{b})) \tag{37}$$

$$\stackrel{(d)}{=} \sum_{t'} \mathbb{P}(y = y | \mathbf{z} = \mathbf{z}, do(\mathbf{b} = \mathbf{b}), t = t') \mathbb{P}(t = t' | \mathbf{z} = \mathbf{z}, do(\mathbf{b} = \mathbf{b})), \tag{38}$$

where $(a)$ follows from Rule 1, (7), and Fact 1, $(b)$ follows from Rule 2, (7), and Fact 1, and $(c)$ follows from Rule 3a and Lemma 2(a), and $(d)$ follows from the law of total probability.

Now, we simplify the first term in (38) as follows:

$$\mathbb{P}(y = y | \mathbf{z} = \mathbf{z}, do(\mathbf{b} = \mathbf{b}), t = t') \stackrel{(a)}{=} \mathbb{P}(y = y | \mathbf{z} = \mathbf{z}, \mathbf{b} = \mathbf{b}, t = t') \tag{39}$$

$$\stackrel{(7)}{=} \mathbb{P}(y = y | \mathbf{z} = \mathbf{z}, t = t'), \tag{40}$$

where $(a)$ follows from Rule 2, (7), and Fact 1. Likewise, we simplify the second term in (38) as follows:

$$\mathbb{P}(t = t' | do(\mathbf{b} = \mathbf{b}), \mathbf{z} = \mathbf{z}) \stackrel{(a)}{=} \mathbb{P}(t = t' | do(\mathbf{b} = \mathbf{b}), \mathbf{z}^{(i)} = \mathbf{z}^{(i)}) \stackrel{(b)}{=} \mathbb{P}(t = t' | \mathbf{z}^{(i)} = \mathbf{z}^{(i)}) \tag{41}$$

$$\stackrel{(c)}{=} \mathbb{P}(t = t'), \tag{42}$$

where $(a)$ follows from Rule 1, (8), and Fact 1, $(b)$ follows from Rule 3a and Lemma 2(d), and $(c)$ follows (8).

Putting together (38), (40), and (42) results in (33).

**Proof of (34):** From the law of total probability, we have

$$\mathbb{P}(\mathbf{z} = \mathbf{z}|do(t = t)) = \sum_b \mathbb{P}(\mathbf{z} = \mathbf{z}|do(t = t), \mathbf{b} = \mathbf{b})\mathbb{P}(\mathbf{b} = \mathbf{b}|do(t = t)). \tag{43}$$

Now, we simplify the first term in (43) as follows:

$$\mathbb{P}(\mathbf{z} = \mathbf{z}|do(t = t), \mathbf{b} = \mathbf{b}) \tag{44}$$

$$\stackrel{(a)}{=} \mathbb{P}(\mathbf{z}^{(i)} = \mathbf{z}^{(i)}|do(t = t), \mathbf{b} = \mathbf{b}) \cdot \mathbb{P}(\mathbf{z}^{(o)} = \mathbf{z}^{(o)}|do(t = t), \mathbf{b} = \mathbf{b}, \mathbf{z}^{(i)} = \mathbf{z}^{(i)}) \tag{45}$$

$$\stackrel{(b)}{=} \mathbb{P}(\mathbf{z}^{(i)} = \mathbf{z}^{(i)}|do(t = t), \mathbf{b} = \mathbf{b}) \cdot \mathbb{P}(\mathbf{z}^{(o)} = \mathbf{z}^{(o)}|t = t, \mathbf{b} = \mathbf{b}, \mathbf{z}^{(i)} = \mathbf{z}^{(i)}) \tag{46}$$

$$\stackrel{(c)}{=} \mathbb{P}(\mathbf{z}^{(i)} = \mathbf{z}^{(i)}|t = t, \mathbf{b} = \mathbf{b}) \cdot \mathbb{P}(\mathbf{z}^{(o)} = \mathbf{z}^{(o)}|t = t, \mathbf{b} = \mathbf{b}, \mathbf{z}^{(i)} = \mathbf{z}^{(i)}) \tag{47}$$

$$\stackrel{(d)}{=} \mathbb{P}(\mathbf{z} = \mathbf{z}|t = t, \mathbf{b} = \mathbf{b}), \tag{48}$$

where $(a)$ and $(d)$ follow from the definition of conditional probability, $(b)$ follows from Rule 2, (8), and Fact 1, and $(c)$ follows from Rule 2 and Lemma 2(c). Likewise, we simplify the second term in (43) as follows:

$$\mathbb{P}(\mathbf{b} = \mathbf{b}|do(t = t)) \stackrel{(a)}{=} \mathbb{P}(\mathbf{b} = \mathbf{b}|t = t), \tag{49}$$

where $(a)$ follows from Rule 2 and Lemma 2(b).

Putting together (43), (48), and (49), results in (34) as follows:

$$\mathbb{P}(\mathbf{z} = \mathbf{z}|do(t = t)) = \sum_b \mathbb{P}(\mathbf{z} = \mathbf{z}|t = t, \mathbf{b} = \mathbf{b})\mathbb{P}(\mathbf{b} = \mathbf{b}|t = t) \stackrel{(a)}{=} \mathbb{P}(\mathbf{z} = \mathbf{z}|t = t), \tag{50}$$

where $(a)$ follows from the law of total probability.

### E.2 Proof of (10)

First, using the law of total probability, we have

$$\mathbb{P}(y = y|do(t = t)) = \sum_b \mathbb{P}(y = y|do(t = t), \mathbf{b} = \mathbf{b})\mathbb{P}(\mathbf{b} = \mathbf{b}|do(t = t)). \tag{51}$$

Now, we show that the first term in RHS of (51) can be simplified as follows

$$\mathbb{P}(y = y|do(t = t), \mathbf{b} = \mathbf{b}) \tag{52}$$

$$= \sum_{\mathbf{z}^{(i)}} \left( \sum_{t'} \mathbb{P}(y = y|\mathbf{s} = \mathbf{s}, t = t')\mathbb{P}(t = t') \right) \mathbb{P}(\mathbf{z}^{(i)} = \mathbf{z}^{(i)}|\mathbf{b} = \mathbf{b}, t = t). \tag{53}$$

where $\mathbf{s} \triangleq (\mathbf{b}, \mathbf{z}^{(i)})$. Using (49) and (53) in (51), completes the proof of (10) as follows:

$$\mathbb{P}(y = y|do(t = t)) \tag{54}$$

$$= \sum_s \left( \sum_{t'} \mathbb{P}(y = y|\mathbf{s} = \mathbf{s}, t = t')\mathbb{P}(t = t') \right) \mathbb{P}(\mathbf{z}^{(i)} = \mathbf{z}^{(i)}|\mathbf{b} = \mathbf{b}, t = t)\mathbb{P}(\mathbf{b} = \mathbf{b}|t = t) \tag{55}$$

$$\stackrel{(a)}{=} \sum_s \left( \sum_{t'} \mathbb{P}(y = y|\mathbf{s} = \mathbf{s}, t = t')\mathbb{P}(t = t') \right) \mathbb{P}(\mathbf{s} = \mathbf{s}|t = t), \tag{56}$$

where $(a)$ follows from the definition of conditional probability.

**Proof of (53):** From the law of total probability, we have

$$\mathbb{P}(y = y|do(t = t), \mathbf{b} = \boldsymbol{b}) \tag{57}$$

$$= \sum_{\boldsymbol{z}^{(i)}} \mathbb{P}(y = y|\mathbf{b} = \boldsymbol{b}, \mathbf{z}^{(i)} = \boldsymbol{z}^{(i)}, do(t = t))\mathbb{P}(\mathbf{z}^{(i)} = \boldsymbol{z}^{(i)}|\mathbf{b} = \boldsymbol{b}, do(t = t)). \tag{58}$$

Now, we simplify the first term in (58) as follows:

$$\mathbb{P}(y = y|\mathbf{b} = \boldsymbol{b}, \mathbf{z}^{(i)} = \boldsymbol{z}^{(i)}, do(t = t)) \tag{59}$$

$$\stackrel{(a)}{=} \mathbb{P}(y = y|do(\mathbf{b} = \boldsymbol{b}), \mathbf{z}^{(i)} = \boldsymbol{z}^{(i)}, do(t = t)) \tag{60}$$

$$\stackrel{(b)}{=} \mathbb{P}(y = y|do(\mathbf{b} = \boldsymbol{b}), \mathbf{z}^{(i)} = \boldsymbol{z}^{(i)}) \tag{61}$$

$$\stackrel{(c)}{=} \sum_{t'} \mathbb{P}(y = y|do(\mathbf{b} = \boldsymbol{b}), \mathbf{z}^{(i)} = \boldsymbol{z}^{(i)}, t = t')\mathbb{P}(t = t'|do(\mathbf{b} = \boldsymbol{b}), \mathbf{z}^{(i)} = \boldsymbol{z}^{(i)}), \tag{62}$$

where $(a)$ follows from Rule 2, Lemma 2(e), and Fact 1, $(b)$ follows from Rule 3a and Lemma 2(a), and $(c)$ follows from the law of total probability. We further simplify the first term in (62) as follows:

$$\mathbb{P}(y = y|do(\mathbf{b} = \boldsymbol{b}), \mathbf{z}^{(i)} = \boldsymbol{z}^{(i)}, t = t') \stackrel{(a)}{=} \mathbb{P}(y = y|\mathbf{b} = \boldsymbol{b}, \mathbf{z}^{(i)} = \boldsymbol{z}^{(i)}, t = t'), \tag{63}$$

where $(a)$ follows from Rule 2 and Lemma 2(e). Using (63) and (42) in (62), we have

$$\mathbb{P}(y = y|\mathbf{b} = \boldsymbol{b}, \mathbf{z}^{(i)} = \boldsymbol{z}^{(i)}, do(t = t)) = \sum_{t'} \mathbb{P}(y = y|\mathbf{b} = \boldsymbol{b}, \mathbf{z}^{(i)} = \boldsymbol{z}^{(i)}, t = t')\mathbb{P}(t = t'). \tag{64}$$

Now, we simplify the second term in (58) as follows:

$$\mathbb{P}(\mathbf{z}^{(i)} = \boldsymbol{z}^{(i)}|\mathbf{b} = \boldsymbol{b}, do(t = t)) \stackrel{(a)}{=} \mathbb{P}(\mathbf{z}^{(i)} = \boldsymbol{z}^{(i)}|\mathbf{b} = \boldsymbol{b}, t = t), \tag{65}$$

where $(a)$ follows from Rule 2 and Lemma 2(c).

Putting together (58), (64), and (65) results in (53).

### E.3 Necessity of Assumption 2

In this section, we provide an example to signify the importance of Assumption 2 to Theorem 3.2. Consider the semi-Markovian causal model in Figure 7 where Assumptions 1 and 3 hold but Assumption 2 does not hold.

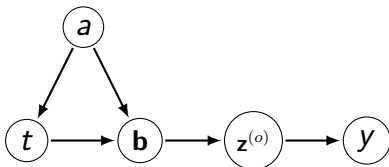

Figure 7: An SMCM signifying the importance of Assumption 2

While $\mathbf{z} = (\mathbf{z}^{(i)}, \mathbf{z}^{(o)})$ satisfies (7) and (8) where $\mathbf{z}^{(i)} = \emptyset$, the causal effect is not equal to the formulae in (9) or (10). To see this, we note that the set $\{a\}$ is a back-door set in Figure 7 implying

$$\mathbb{P}(y|do(t = t)) = \sum_a \mathbb{P}(y|a, t)\mathbb{P}(a). \tag{66}$$

Now, we simplify the right hand side of (66) to show explicitly that it is not equivalent to (9). From the law of total probability, we have

$$\mathbb{P}(y|a, t) = \sum_{\mathbf{z}} \mathbb{P}(y|\mathbf{z}, a, t)\mathbb{P}(\mathbf{z}|a, t) \tag{67}$$

$$\stackrel{(a)}{=} \sum_{\mathbf{z}} \Big( \sum_{t'} \mathbb{P}(y|\mathbf{z}, a, t)\mathbb{P}(t') \Big) \mathbb{P}(\mathbf{z}|a, t) \stackrel{(b)}{=} \sum_{\mathbf{z}} \Big( \sum_{t'} \mathbb{P}(y|\mathbf{z}, t')\mathbb{P}(t') \Big) \mathbb{P}(\mathbf{z}|a, t), \quad (68)$$

where $(a)$ follows because $\sum_{t'} \mathbb{P}(t') = 1$ and $(b)$ follows $y$ is independent of every other variable conditioned on $\mathbf{z}$. Plugging (68) in (66), we have

$$\mathbb{P}(y|do(t = t)) = \sum_{\mathbf{z}} \Big( \sum_{t'} \mathbb{P}(y|\mathbf{z}, t')\mathbb{P}(t') \Big) \Big( \sum_{a} \mathbb{P}(\mathbf{z}|a, t)\mathbb{P}(a) \Big). \quad (69)$$

Lastly, using the law of total probability, (9) can be rewritten as

$$\mathbb{P}(y|do(t = t)) = \sum_{\mathbf{z}} \Big( \sum_{t'} \mathbb{P}(y|\mathbf{z}, t')\mathbb{P}(t') \Big) \Big( \sum_{a} \mathbb{P}(\mathbf{z}|a, t)\mathbb{P}(a|t) \Big). \quad (70)$$

Therefore, the variables $a$ and $t$ could be such that (69) is different from (70). We note that similar steps can be used to show that (66) is not equivalent to (10). In conclusion, Assumption 2 is crucial for the formulae in (9) and (10) to hold.

# F   Proof of Lemma 2

First, we state the following d-separation criterion used to prove Lemma 2(b) and Lemma 2(d). See Appendix F.1 for a proof.

**Lemma 3.** *Suppose Assumptions 1 to 3 hold. Then, $t \perp\!\!\!\perp_d \mathbf{b}|\mathbf{z}^{(i)}$ in $\mathcal{G}_{\underline{t}}$.*

Now, we prove each part of Lemma 2 one-by-one.

**Proof of Lemma 2(a)**   In $\mathcal{G}_{\overline{\mathbf{b}}}^{t}$, all edges going into $\mathbf{b}$ are removed. Under Assumption 3, this implies that all edges going out of $t$ are removed. Now, consider any path $\mathcal{P}(F_t, y)$ between $F_t$ and $y$ in $\mathcal{G}_{\overline{\mathbf{b}}}^{t}$. This path takes one of the following two forms: $(a)$ $F_t \longrightarrow t \longleftarrow \cdots y$ or $(b)$ $F_t \longrightarrow t \longleftrightarrow \cdots y$. In either case, there is a collider at $t$ in $\mathcal{P}(F_t, y)$. This collider is blocked when $\mathbf{z}$ and $\mathbf{b}$ are conditioned on because $t \notin \mathbf{z}$, $t \notin \mathbf{b}$, and $t$ does not have any descendants in $\mathcal{G}_{\overline{\mathbf{b}}}^{t}$. Therefore, $y \perp\!\!\!\perp_d F_t|\mathbf{z}, \mathbf{b}$ in $\mathcal{G}_{\overline{\mathbf{b}}}^{t}$. Similarly, the collider is blocked when $\mathbf{z}^{(i)}$ and $\mathbf{b}$ are conditioned on because $t \notin \mathbf{z}^{(i)}$, $t \notin \mathbf{b}$, and $t$ does not have any descendants in $\mathcal{G}_{\overline{\mathbf{b}}}^{t}$. Therefore, $y \perp\!\!\!\perp_d F_t|\mathbf{z}^{(i)}, \mathbf{b}$ in $\mathcal{G}_{\overline{\mathbf{b}}}^{t}$.

**Proof of Lemma 2(b)**   We prove this by contradiction. Assume there exists at least one unblocked path between $t$ and some $b \in \mathbf{b}$ in $\mathcal{G}_{\underline{t}}$. Let $\mathcal{P}(t, b)$ denote any such unblocked path.

Suppose we condition on $\mathbf{z}^{(i)}$. From Lemma 3, $\mathcal{P}(t, b)$ is blocked in $\mathcal{G}_{\underline{t}}$ when $\mathbf{z}^{(i)}$ is conditioned on. Let $v$ be any node at which $\mathcal{P}(t, b)$ is blocked in $\mathcal{G}_{\underline{t}}$ when $\mathbf{z}^{(i)}$ is conditioned on. We must have that $v \in \mathcal{P}(t, b) \setminus \{t, b\}$ and $v \in \mathbf{z}^{(i)}$. Then, the path $\mathcal{P}(t, v) \subset \mathcal{P}(t, b)$ is unblocked in $\mathcal{G}_{\underline{t}}$ when $\mathbf{z}^{(i)}$ is unconditioned on. However, this contradicts $t \perp\!\!\!\perp_d \mathbf{z}^{(i)}$ in $\mathcal{G}_{\underline{t}}$ (which follows from (8)$(i)$ and Fact 1).

**Proof of Lemma 2(c)**   We prove this by contradiction. Assume there exists at least one unblocked path between $t$ and some $z^{(i)} \in \mathbf{z}^{(i)}$ in $\mathcal{G}_{\underline{t}}$ when $\mathbf{b}$ is conditioned on. Let $\mathcal{P}(t, z^{(i)})$ denote any such unblocked path.

Suppose, we uncondition on $\mathbf{b}$. From (8)(i) and Fact 1, we have $t \perp\!\!\!\perp_d \mathbf{z}^{(i)}$ in $\mathcal{G}_{\underline{t}}$. Therefore, $\mathcal{P}(t, z^{(i)})$ is blocked in $\mathcal{G}_{\underline{t}}$ when $\mathbf{b}$ is unconditioned on. Now, we create a set $\mathbf{v}$ consisting of all the nodes at which $\mathcal{P}(t, z^{(i)})$ is blocked in $\mathcal{G}_{\underline{t}}$ when $\mathbf{b}$ is unconditioned on. Define the set $\mathbf{v}$ such that for any $v \in \mathbf{v}$, the following are true: $(a)$ $v \in \mathcal{P}(t, z^{(i)}) \setminus \{t, z^{(i)}\}$, $(b)$ $\mathcal{P}(t, z^{(i)})$ contains a collider at $v$ in $\mathcal{G}_{\underline{t}}$, and $(c)$ there exists an unblocked descendant path from $v$ to some $b \in \mathbf{b}$ in $\mathcal{G}_{\underline{t}}$.

Now, we must have $\mathbf{v} \neq \emptyset$, since $\mathcal{P}(t, z^{(i)})$ is blocked in $\mathcal{G}_{\underline{t}}$ when $\mathbf{b}$ is unconditioned on. Let $v_c \in \mathbf{v}$ be that node which is closest to $t$ in the path $\mathcal{P}(t, z^{(i)})$, and let $\mathcal{P}(v_c, b)$ be an unblocked descendant path from $v$ to some $b \in \mathbf{b}$ in $\mathcal{G}_{\underline{t}}$ (there must be one from the definition of the set $\mathbf{v}$). Consider the path $\mathcal{P}(t, b)$ obtained after concatenating $\mathcal{P}(t, v_c) \subset \mathcal{P}(t, z^{(i)})$ and $\mathcal{P}(v_c, b)$. By the definition of $\mathbf{v}$ and the choice of $v_c$, $\mathcal{P}(t, b)$ is unblocked in $\mathcal{G}_{\underline{t}}$ since $(a)$ $\mathcal{P}(t, v_c)$ is unblocked in $\mathcal{G}_{\underline{t}}$, $(b)$ $\mathcal{P}(v_c, b)$ is unblocked in $\mathcal{G}_{\underline{t}}$, and $(c)$ there is no collider at $v_c$ in $\mathcal{P}(t, b)$. However, this contradicts $t \perp\!\!\!\perp_d \mathbf{b}$ in $\mathcal{G}_{\underline{t}}$ (which follows from Lemma 2(b)).

**Proof of Lemma 2(d)**  We prove this by contradiction. Assume there exists at least one unblocked path between $t$ and $F_{\mathbf{b}}$ in $\mathcal{G}^{\mathbf{b}}$ when $\mathbf{z}^{(i)}$ is conditioned on. Let $\mathcal{P}(t, F_{\mathbf{b}})$ denote the shortest of these unblocked path. By definition of $\mathcal{G}^{\mathbf{b}}$, this path has to be of the form: $t \cdots , b \leftarrow F_{\mathbf{b}}$ for some $b \in \mathbf{b}$. Now, we have the following three cases:

    (i) $\mathcal{P}(t, F_{\mathbf{b}})$ contains $t \longrightarrow b$: In this case, because a path is a sequence of distinct nodes, $\mathcal{P}(t, F_{\mathbf{b}})$ has to be $t \longrightarrow b \leftarrow F_{\mathbf{b}}$. By assumption, $\mathcal{P}(t, F_{\mathbf{b}})$ is unblocked when $\mathbf{z}^{(i)}$ is conditioned on. Since there is a collider at $b$ in $\mathcal{P}(t, F_{\mathbf{b}})$, there exists at least one unblocked descendant path from $b$ to $\mathbf{z}^{(i)}$ when $\mathbf{z}^{(i)}$ is conditioned on. Let $\mathcal{P}(b, z^{(i)})$ denote the shortest of these paths from $b$ to some $z^{(i)} \in \mathbf{z}^{(i)}$ in $\mathcal{G}^{\mathbf{b}}$. We note that this path also exists in $\mathcal{G}$ and is of the form $b \longrightarrow \cdots \longrightarrow z^{(i)}$

    Suppose we uncondition on $\mathbf{z}^{(i)}$. Consider the path $\mathcal{P}(t, z^{(i)}) \supset \mathcal{P}(b, z^{(i)})$ between $t$ and $z^{(i)}$ of the form $t \longrightarrow b \longrightarrow \cdots \longrightarrow z^{(i)}$ in $\mathcal{G}$. This path remains unblocked even when $\mathbf{z}^{(i)}$ is unconditioned on as it does not have any colliders. This contradicts $z^{(i)} \perp\!\!\!\perp_d t$ (which follows from (8)).

    (ii) $\mathcal{P}(t, F_{\mathbf{b}})$ contains $t \longrightarrow b_1$ for some $b_1 \in \mathbf{b}$ such that $b_1 \neq b$: In this case, the path $\mathcal{P}(t, F_{\mathbf{b}})$ has to be of the form $t \longrightarrow b_1 \cdots b \leftarrow F_{\mathbf{b}}$. Therefore, there exists at least one collider on the path $\mathcal{P}(t, F_{\mathbf{b}})$. Let $v \in \mathcal{P}(t, F_{\mathbf{b}}) \setminus \{t, F_{\mathbf{b}}\}$ be the collider on the path $\mathcal{P}(t, F_{\mathbf{b}})$ that is closest to $b_1$. Consider the path $\mathcal{P}(t, v) \subset \mathcal{P}(t, F_{\mathbf{b}})$. We note that this path also exists in $\mathcal{G}$ and is of the form $t \longrightarrow b_1 \longrightarrow \cdots \longrightarrow v$.

    By assumption, $\mathcal{P}(t, F_{\mathbf{b}})$ is unblocked when $\mathbf{z}^{(i)}$ is conditioned on. Since there is a collider at $v$ in $\mathcal{P}(t, F_{\mathbf{b}})$, there exists at least one unblocked descendant path from $v$ to $\mathbf{z}^{(i)}$ when $\mathbf{z}^{(i)}$ is conditioned on. Let $\mathcal{P}(v, z^{(i)})$ denote the shortest of these paths from $v$ to some $z^{(i)} \in \mathbf{z}^{(i)}$ in $\mathcal{G}^{\mathbf{b}}$. We note that this path also exists in $\mathcal{G}$ and is of the form $v \longrightarrow \cdots \longrightarrow z^{(i)}$.

    Suppose we uncondition on $\mathbf{z}^{(i)}$. Consider the path $\mathcal{P}(t, z^{(i)})$ between $t$ and $z^{(i)}$ in $\mathcal{G}$ obtained after concatenating $\mathcal{P}(t, v) \subset \mathcal{P}(t, F_{\mathbf{b}})$ and $\mathcal{P}(v, z^{(i)})$. This path, of the form $t \longrightarrow b_1 \longrightarrow \cdots \longrightarrow v \longrightarrow \cdots \longrightarrow z^{(i)}$, remains unblocked even when $\mathbf{z}^{(i)}$ is unconditioned on as it does not have any colliders. This contradicts $z^{(i)} \perp\!\!\!\perp_d t$ (which follows from (8)).

    (ii) $\mathcal{P}(t, F_{\mathbf{b}})$ does not contain $t \longrightarrow b_1$ for every $b_1 \in \mathbf{b}$: By assumption, $\mathcal{P}(t, F_{\mathbf{b}})$ is unblocked in $\mathcal{G}^{\mathbf{b}}$ when $\mathbf{z}^{(i)}$ is conditioned on. Therefore, if $\mathcal{P}(t, F_{\mathbf{b}})$ does not contain the edge $t \longrightarrow b_1$ for any $b_1 \in \mathbf{b}$, there exists a path $\mathcal{P}(t, b)$ between $t$ to $b$ in $\mathcal{G}$ that is unblocked when $\mathbf{z}^{(i)}$ is conditioned on, and takes one of the following two forms: $(a)$ $t \leftarrow \cdots b$ or $(b)$ $t \leftrightarrow \cdots b$. Then, it is easy to see that the path $\mathcal{P}(t, b)$ also remains unblocked in $\mathcal{G}_{\underline{t}}$ while $\mathbf{z}^{(i)}$ is conditioned on. However, this contradicts $t \perp\!\!\!\perp_d \mathbf{b} | \mathbf{z}^{(i)}$ in $\mathcal{G}_{\underline{t}}$ (which follows from Lemma 3).

**Proof of Lemma 2(e)**  We prove this by contradiction. Assume there exists at least one unblocked path between $y$ and some $b \in \mathbf{b}$ in $\mathcal{G}_{\underline{\mathbf{b}}}$ when $t$ and $\mathbf{z}^{(i)}$ are conditioned on. Let $\mathcal{P}(b, y)$ denote the shortest of these unblocked path. Therefore, no $b_1 \in \mathbf{b}$, such that $b_1 \neq b$, is on the path $\mathcal{P}(b, y)$, i.e., $b_1 \notin \mathcal{P}(b, y)$. Further, $\mathcal{P}(b, y)$ takes one of the following two forms because all the edges going out of $\mathbf{b}$ are removed in $\mathcal{G}_{\underline{\mathbf{b}}}$: $(a)$ $b \leftarrow \cdots y$ or $(b)$ $b \leftrightarrow \cdots y$.

Suppose we condition on $\mathbf{z}^{(o)}$ (while $t$ and $\mathbf{z}^{(i)}$ are still conditioned on). From (7) and Fact 1, we have $y \perp\!\!\!\perp_d b | t, \mathbf{z}$ in $\mathcal{G}_{\underline{\mathbf{b}}}$. Therefore, the path $\mathcal{P}(b, y)$ is blocked in $\mathcal{G}_{\underline{\mathbf{b}}}$ when $\mathbf{z}^{(o)}$ is conditioned on (while $t$ and $\mathbf{z}^{(i)}$ are still conditioned on). Let $v$ be any node at which $\mathcal{P}(b, y)$ is blocked in $\mathcal{G}_{\underline{\mathbf{b}}}$ when $\mathbf{z}^{(o)}$ is conditioned on (while $t$ and $\mathbf{z}^{(i)}$ are still conditioned on). We must have that $v \in \mathcal{P}(b, y) \setminus \{y, b\}$ and $v \in \mathbf{z}^{(o)}$. Suppose we uncondition on $\mathbf{z}^{(o)}$ (while $t$ and $\mathbf{z}^{(i)}$ are still conditioned on). Then, the path $\mathcal{P}(b, v) \subset \mathcal{P}(b, y)$ is unblocked in $\mathcal{G}_{\underline{\mathbf{b}}}$.

We consider the following two scenarios depending on whether or not $\mathcal{P}(b, v)$ contains $t$. In both scenarios, we show that there is an unblocked path between $t$ and $v$ in $\mathcal{G}_{\underline{\mathbf{b}}}$ when we condition on $\mathbf{b}$ (while $t$ and $\mathbf{z}^{(i)}$ are still conditioned on).

    (i) $\mathcal{P}(b, v)$ contains $t$: Consider the path $\mathcal{P}(t, v) \subset \mathcal{P}(b, v)$ which is unblocked in $\mathcal{G}_{\underline{\mathbf{b}}}$ when $t$ and $\mathbf{z}^{(i)}$ are conditioned on. Further, by the choice of $\mathcal{P}(b, y)$, no $b_1 \in \mathbf{b}$ is on the path

$\mathcal{P}(t, v)$. Therefore, the path $\mathcal{P}(t, v)$ in $\mathcal{G}_{\underline{\mathbf{b}}}$ remains unblocked when we condition on $\mathbf{b}$ (while $t$ and $\mathbf{z}^{(i)}$ are still conditioned on).

(ii) $\mathcal{P}(b, v)$ does not contain $t$: Consider the path $\mathcal{P}(t, v) \supset \mathcal{P}(b, v)$ (by including the extra edge $t \rightarrow b$) which takes one of the following two forms: $(a)$ $t \longrightarrow b \longleftarrow \cdots v$ or $(b)$ $t \longrightarrow b \longleftrightarrow \cdots v$. Further, by the choice of $\mathcal{P}(b, y)$, no $b_1 \in \mathbf{b}$ ($b_1 \neq b$) is on the path $\mathcal{P}(t, v)$. Suppose we condition on $\mathbf{b}$ (while $t$ and $\mathbf{z}^{(i)}$ are still conditioned on). Then, the path $\mathcal{P}(t, v)$ in $\mathcal{G}_{\underline{\mathbf{b}}}$ is unblocked because $(a)$ the collider at $b$ is unblocked when $\mathbf{b}$ is conditioned on and $(b)$ the path $\mathcal{P}(b, v)$ in $\mathcal{G}_{\underline{\mathbf{b}}}$ remains unblocked when $\mathbf{b}$ is conditioned on (while $t$ and $\mathbf{z}^{(i)}$ are still conditioned on).

Now, suppose we uncondition on $t$ (while $\mathbf{b}$ and $\mathbf{z}^{(i)}$ are still conditioned on). We have the following two scenarios depending on whether or not $\mathcal{P}(t, v)$ in $\mathcal{G}_{\underline{\mathbf{b}}}$ remains unblocked. In both scenarios, we show that there is an unblocked path between $t$ and $v$ in $\mathcal{G}_{\underline{\mathbf{b}}}$ when we uncondition on $t$ (while $\mathbf{b}$ and $\mathbf{z}^{(i)}$ are still conditioned on).

1. If $\mathcal{P}(t, v)$ remains unblocked: In this case, $\mathcal{P}(t, v)$ in $\mathcal{G}_{\underline{\mathbf{b}}}$ is an unblocked path between $t$ and $v$ when $\mathbf{z}^{(i)}$ and $\mathbf{b}$ are conditioned on, as desired.

2. If $\mathcal{P}(t, v)$ does not remain unblocked: In this case, it is the unconditioning on $t$ (while $\mathbf{b}$ and $\mathbf{z}^{(i)}$ are still conditioned on) that blocks $\mathcal{P}(t, v)$. Now, we create a set $\mathbf{w}$ consisting of all the nodes at which $\mathcal{P}(t, v)$ is blocked in $\mathcal{G}_{\underline{\mathbf{b}}}$ when $t$ is unconditioned on (while $\mathbf{b}$ and $\mathbf{z}^{(i)}$ are still conditioned on). Define the set $\mathbf{w}$ such that for any $w \in \mathbf{w}$, the following are true: $(a)$ $w \in \mathcal{P}(t, v) \setminus \{t, v\}$, $(b)$ $\mathcal{P}(t, v)$ contains a collider at $w$ in $\mathcal{G}_{\underline{\mathbf{b}}}$, and $(c)$ there exists an unblocked descendant path from $w$ to $t$ in $\mathcal{G}_{\underline{\mathbf{b}}}$.

Now, we must have $\mathbf{w} \neq \emptyset$, since $\mathcal{P}(t, v)$ is blocked in $\mathcal{G}_{\underline{\mathbf{b}}}$ when $t$ is unconditioned on (while $\mathbf{b}$ and $\mathbf{z}^{(i)}$ are still conditioned on). Let $w_c \in \mathbf{w}$ be that node which is closest to $v$ in the path $\mathcal{P}(t, v)$, and let $\mathcal{P}(w_c, t)$ be an unblocked descendant path from $w_c$ to $t$ in $\mathcal{G}_{\underline{\mathbf{b}}}$ (there must be one from the definition of the set $\mathbf{w}$). Consider the path $\mathcal{P}'(v, t)$ obtained after concatenating $\mathcal{P}(v, w_c) \subset \mathcal{P}(t, v)$ and $\mathcal{P}(w_c, t)$. By the definition of $\mathbf{w}$ and the choice of $w_c$, $\mathcal{P}'(v, t)$ is unblocked in $\mathcal{G}_{\underline{\mathbf{b}}}$ when $t$ is unconditioned on (while $\mathbf{b}$ and $\mathbf{z}^{(i)}$ are still conditioned on) since $(a)$ $\mathcal{P}(v, w_c)$ is unblocked, $(b)$ $\mathcal{P}(w_c, t)$ is unblocked, and $(c)$ there is no collider at $w_c$ in $\mathcal{P}'(v, t)$. Therefore, we have an unblocked path between $t$ and $v$ in $\mathcal{G}_{\underline{\mathbf{b}}}$ when $\mathbf{z}^{(i)}$ and $\mathbf{b}$ are conditioned on, as desired.

To conclude the proof, we note that the existence of an unblocked path between $t$ and $v \in \mathbf{z}^{(o)}$ in $\mathcal{G}_{\underline{\mathbf{b}}}$ when $\mathbf{z}^{(i)}$ and $\mathbf{b}$ are conditioned on contradicts $\mathbf{z}^{(o)} \perp\!\!\!\perp_d t | \mathbf{b}, \mathbf{z}^{(i)}$ in $\mathcal{G}_{\underline{\mathbf{b}}}$ (which follows from (8) and Fact 1).

### F.1 Proof of Lemma 3

First, we claim $t \perp\!\!\!\perp_d \mathbf{b} | \mathbf{z}$ in $\mathcal{G}_{\underline{t}}$. We assume this claim and proceed to prove the statement in the Lemma by contradiction. Assume there exists at least one unblocked path between $t$ and some $b \in \mathbf{b}$ in $\mathcal{G}_{\underline{t}}$ when $\mathbf{z}^{(i)}$ is conditioned on. Let $\mathcal{P}(t, b)$ denote the shortest of these unblocked path. Therefore, no $b_1 \in \mathbf{b}$ such that $b_1 \neq b$ is not on the path $\mathcal{P}(t, b)$, i.e., $b_1 \notin \mathcal{P}(t, b)$.

Suppose we condition on $\mathbf{z}^{(o)}$ (while $\mathbf{z}^{(i)}$ is still conditioned on). From the claim, $\mathcal{P}(t, b)$ is blocked in $\mathcal{G}_{\underline{t}}$ when $\mathbf{z}^{(o)}$ is conditioned on (while $\mathbf{z}^{(i)}$ is still conditioned on). Let $v$ be any node at which $\mathcal{P}(t, b)$ is blocked in $\mathcal{G}_{\underline{t}}$ when $\mathbf{z}^{(o)}$ is conditioned on (while $\mathbf{z}^{(i)}$ is still conditioned on). We must have that $v \in \mathcal{P}(t, b) \setminus \{t, b\}$ and $v \in \mathbf{z}^{(o)}$. Then, the path $\mathcal{P}(t, v) \subset \mathcal{P}(t, b)$ is unblocked in $\mathcal{G}_{\underline{t}}$ when $\mathbf{z}^{(o)}$ is unconditioned on (while $\mathbf{z}^{(i)}$ is still conditioned on). Further, no $b \in \mathbf{b}$ is on the path $\mathcal{P}(t, v)$. As a result, the path $\mathcal{P}(t, v)$ remains unblocked when $\mathbf{b}$ is conditioned on (while $\mathbf{z}^{(i)}$ is still conditioned on). However, this contradicts $t \perp\!\!\!\perp_d \mathbf{z}^{(o)} | b, \mathbf{z}^{(i)}$ in $\mathcal{G}_{\underline{t}}$ (which follows from (8)$(ii)$ and Fact 1).

*Proof of Claim - $t \perp\!\!\!\perp_d \mathbf{b} | \mathbf{z}$ in $\mathcal{G}_{\underline{t}}$:* It remains to prove the claim $t \perp\!\!\!\perp_d \mathbf{b} | \mathbf{z}$ in $\mathcal{G}_{\underline{t}}$. We prove this by contradiction. Assume there exists at least one unblocked path between $t$ and some $b \in \mathbf{b}$ in $\mathcal{G}_{\underline{t}}$ when $\mathbf{z}$ is conditioned on. Let $\mathcal{P}(t, b)$ denote any such unblocked path. This path takes one of the following two forms: $(a)$ $t \longleftarrow \cdots b$ or $(b)$ $t \longleftrightarrow \cdots b$ because all edges going out of $t$ are removed in $\mathcal{G}_{\underline{t}}$.

Suppose we condition on $t$ (while $\mathbf{z}$ is still conditioned on). The path $\mathcal{P}(t, b)$ remains unblocked because $t \notin \mathcal{P}(t, b)$ (a path is a sequence of distinct nodes). Then, the path $\mathcal{P}(y, b) \supset \mathcal{P}(t, b)$ of the form $(a)$ $y \leftrightarrow t \leftarrow \cdots b$ or $(b)$ $y \leftrightarrow t \leftrightarrow \cdots b$ is unblocked because the additional conditioning on $t$ (while $\mathbf{z}$ is still conditioned on) unblocks the collider at $t$. However, this contradicts $\mathbf{b} \perp\!\!\!\perp_d y | t, \mathbf{z}$ in $\mathcal{G}_{\underline{t}}$ (which follows from (7) and Fact 1).

# G  Experimental Results

In this section, we provide additional experimental results. First, we provide more details regarding the numerical example in Section 3.1. Next, we demonstrate the applicability of our method on a class of graphs slightly different from the one in Section 4.1. Then, we provide the 6 random graphs from Section 4.2 as well as ATE estimation results on specific choices of SMCMs including the one in Figure 2. Finally, we provide histograms analogous to Figure 6 for the second choice of $\mathbf{b}$ on German credit dataset as well as details about our analysis with Adult dataset.

## G.1  Numerical example in Section 3.1

The observed variables for this example also follow the structural equation model in (11). Also, to generate the true ATE, we intervene on the generation model in (11) by setting $t = 0$ and $t = 1$.

## G.2  Applicability to a class of random graphs

As in Section 4.1, we create a class of random SMCMs, sample 100 SMCMs from this class, and check if (7) and (8) hold by checking for corresponding d-separations in the SMCMs. The class of random graphs considered here is analogous to the class of random graphs considered in Section 4.1 expect for the choice of $t$. Here, we choose any variable that is ancestor of $y$ but not its parent as $t$. This is in contrast to Section 4.1 where we choose any variable that is ancestor of $y$ but not its parent or grandparent as $t$. We compare the success rate of the same two approaches: $(i)$ exhaustive search for $\mathbf{z}$ satisfying (7) and (8) and $(ii)$ search for a $\mathbf{z}$ of size at-most 5 satisfying (7) and (8). We provide the number of successes of these approaches as a tuple in Table 2 for various $p$, $d$, and $q$. As before, we see that the two approaches have comparable performances and the IDP algorithm gives 0 successes across various $p$, $d$, and $q$ even though it is supplied with the true PAG. Also, as expected the number of successes for this class of graphs is much lower than the class considered in Section 4.1.

Table 2: Number of successes out of 100 random graphs for our methods shown as a tuple. The first method searches a $\mathbf{z}$ exhaustively and the second method searches a $\mathbf{z}$ with size at-most 5.

|  | $p = 10$ | | | $p = 15$ | | |
|  | $d = 2$ | $d = 3$ | $d = 4$ | $d = 2$ | $d = 3$ | $d = 4$ |
| --- | --- | --- | --- | --- | --- | --- |
| $q = 0.0$ | $(6, 6)$ | $(3, 3)$ | $(1, 1)$ | $(11, 11)$ | $(2, 2)$ | $(1, 1)$ |
| $q = 0.5$ | $(3, 3)$ | $(0, 0)$ | $(0, 0)$ | $(5, 5)$ | $(2, 2)$ | $(1, 1)$ |
| $q = 1.0$ | $(1, 1)$ | $(0, 0)$ | $(0, 0)$ | $(1, 1)$ | $(0, 0)$ | $(0, 0)$ |

## G.3  ATE estimation

We also conduct ATE estimation experiments on four specific SMCMs. The first SMCM is the graph $\mathcal{G}^{toy}$ in Figure 2. The remaining graphs, named $\mathcal{G}_i^{toy}$, $i \in \{1, 2, 3\}$, are shown in Figure 8, and are obtained by adding additional edges and modifying $\mathcal{G}^{toy}$. These SMCMs are designed in a way such that there exists $\mathbf{z} = (\mathbf{z}^{(i)}, \mathbf{z}^{(o)})$ satisfying the conditional independence statements in Theorem 3.2.

We follow a data generation procedure similar to the one in Section 4.2. In contrast, we show the performance of our approach for a fixed $n$ but different thresholds of p-value $p_v$. We average the ATE error over 50 runs where in each run we set $n = 50000$. As we see in Figure 9, both the ATE estimates returned by Algorithm 1 are far superior compared to the naive front-door adjustment using $\mathbf{b}$.

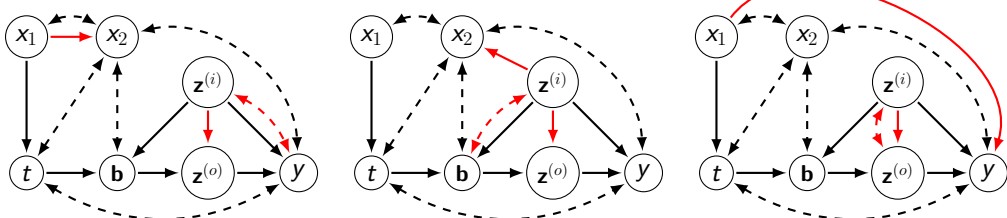

Figure 8: The causal graphs used to further validate our theoretical results. These are obtained by adding additional edges (shown in red) to $\mathcal{G}^{toy}$ in Figure 2. We denote these graphs (from left to right) by $\mathcal{G}_1^{toy}$, $\mathcal{G}_2^{toy}$, and $\mathcal{G}_3^{toy}$, respectively.

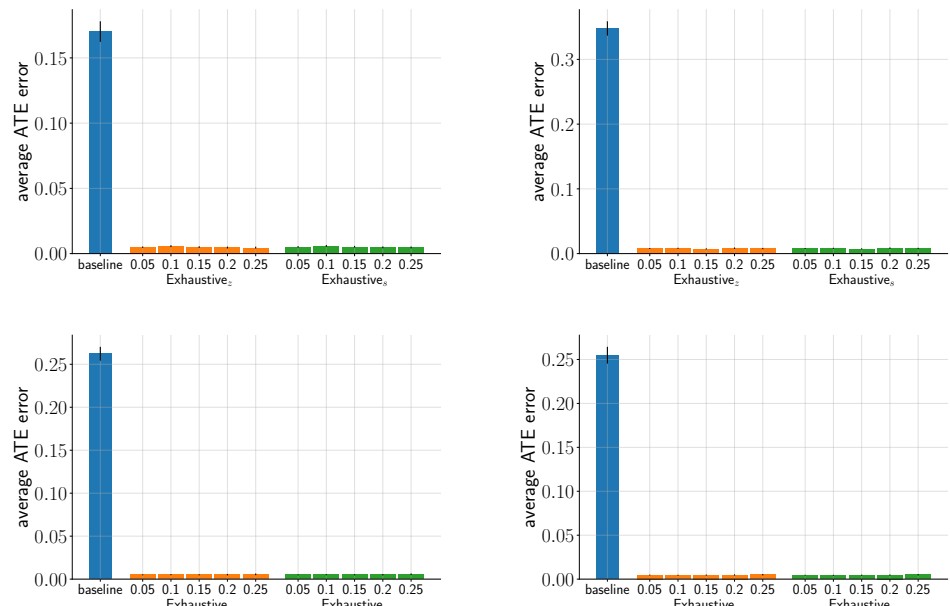

Figure 9: Performance of Algorithm 1 for different p-value thresholds $p_v$ on $\mathcal{G}^{toy}$ in Figure 2 on top left, on $\mathcal{G}_1^{toy}$ from Figure 8 on top right, on $\mathcal{G}_2^{toy}$ in Figure 8 on bottom left, and on $\mathcal{G}_3^{toy}$ from Figure 8 on bottom right

In Figure 11, we provide the 6 random SMCMs used in Section 4.2. As mentioned in Section 4.1, we choose the last variable in the causal ordering as $y$ and a variable that is ancestor of $y$ but not its parent or grandparent as $t$. We also show the corresponding $\mathbf{z} = (\mathbf{z}^{(i)}, \mathbf{z}^{(o)})$ satisfying (7) and (8).

### G.4 German Credit dataset

As in Section 4.3, we assess the conditional independence associated with the selected $\mathbf{z}$ for the choice of $\mathbf{b} = \{\#$ of people financially dependent on the applicant, applicant's savings$\}$, Algorithm 1 results in $\mathbf{z}^{(i)} = \{$purpose for which the credit was needed, applicant's checking account status with the bank$\}$ via 100 random bootstraps. We show the corresponding p-values for these bootstraps in a histogram in Figure 10 below. As expected, we observe the p-values to be spread out.

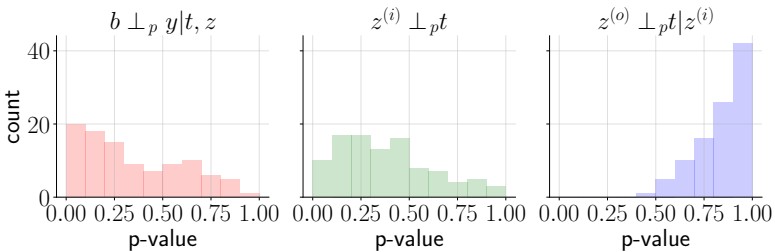

Figure 10: Histograms of p-values of the conditional independencies in (9) and (10) over 100 bootstrap runs for $\mathbf{b} = \{$# of people financially dependent on the applicant, applicant's savings$\}$, Algorithm 1 results in $\mathbf{z}^{(i)} = \{$purpose for which the credit was needed, applicant's checking account status with the bank$\}$.

## G.5 Adult dataset

The Adult dataset [Kohavi and Becker, 1996] is used for income analysis where the goal is to predict whether an individual's income is more than \$50,000 using 14 demographic and socio-economic features. The sensitive attribute $t$ is the individual's sex, either male or female. Further, the categorical attributes are one-hot encoded. As with German Credit dataset, we apply Algorithm 1 with $n_r = 100$ and $p_v = 0.1$ where we search for a set $\mathbf{z} = (\mathbf{z}^{(o)}, \mathbf{z}^{(i)})$ of size at most 3 under the following two assumptions on the set of all children $\mathbf{b}$ of $t$: (1) $\mathbf{b} = \{$# individual's relationship status (which includes wife/husband)$\}$ and (2) $\mathbf{b} = \{$# individual's relationship status (which includes wife/husband), individual's occupation$\}$. In either case, Algorithm 1 was unable to find a suitable $\mathbf{z}$ satisfying $\mathbf{b} \perp_p y|\mathbf{z}, t$. This suggests that in this dataset, there may not be any non-child descendants of the sensitive attribute, which is required for our criterion to hold.

## G.6 Licenses

In this work, we used a workstation with an AMD Ryzen Threadripper 3990X 64-Core Processor (128 threads in total) with 256 GB RAM and 2x Nvidia RTX 3090 GPUs. However, our simulations only used the CPU resources of the workstation.

We mainly relied on the following Python repositories — (a) networkx (`https://networkx.org`), (b) causal-learn (`https://causal-learn.readthedocs.io/en/latest/`), (c) RCoT [Strobl et al., 2019b] and (d) ridgeCV, (`https://github.com/scikit-learn/scikit-learn/tree/15a949460/sklearn/linear_model/_ridge.py`). We did not modify any of the code under licenses; we only installed these repositories as packages.

In addition to these, we used two public datasets (a) German Credit dataset (`https://archive.ics.uci.edu/ml/datasets/statlog+(german+credit+data)`) and (b) Adult dataset (`https://archive.ics.uci.edu/ml/datasets/adult`). These datasets are commonly used benchmark datasets for causal fairness, which is why we chose them for our comparisons.

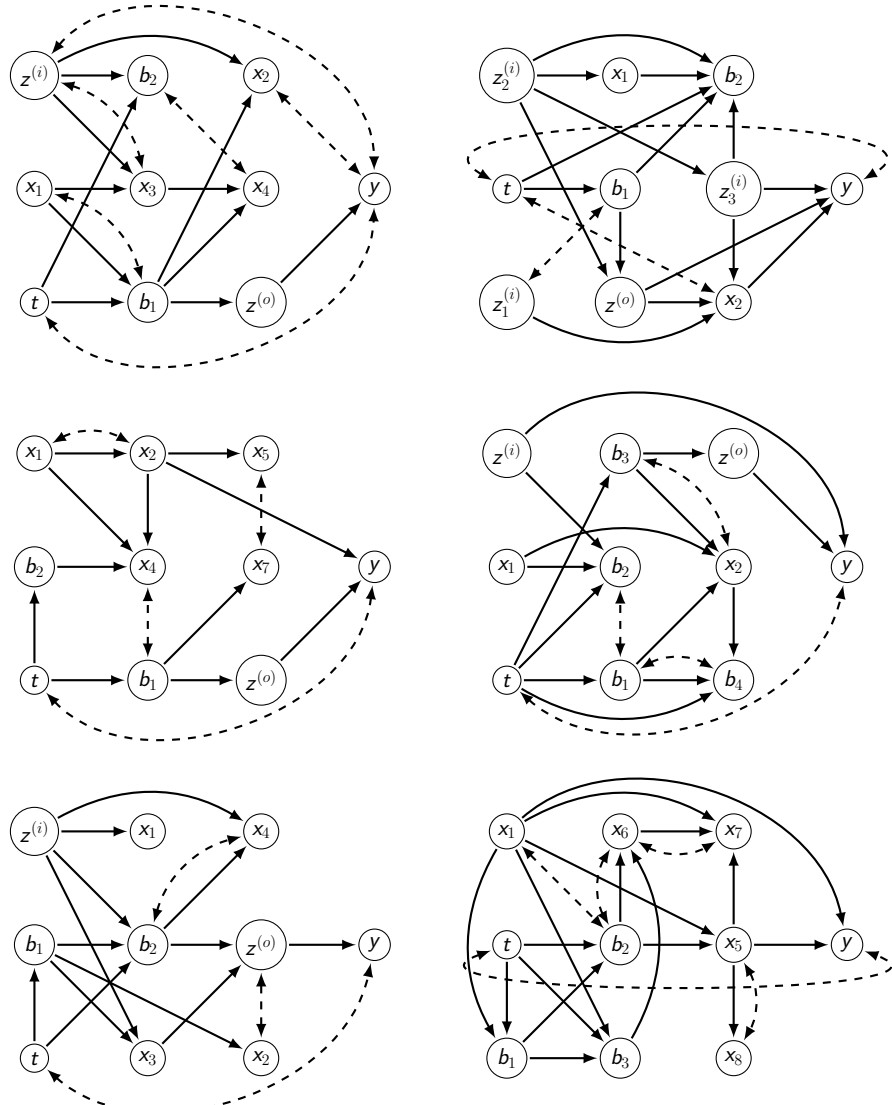

Figure 11: The SMCMs used in Section 4.2 to compare Algorithm 1 with the `Baseline` that uses **b** for front-door adjustment. These are the 6 out of the 100 random graphs in Section 4.1 for $p = 10$, $d = 2$, and $q = 1.0$ where our approach was successful indicating existence of $\mathbf{z} = (\mathbf{z}^{(i)}, \mathbf{z}^{(o)})$ such that the conditional independence statements in Theorem 3.2 hold.

