# OpenReview forum: "Front-door Adjustment Beyond Markov Equivalence with Limited Graph Knowledge"
_NeurIPS.cc/2023/Conference — NeurIPS 2023 poster_

### Official Review · Reviewer_XvUT · 2023-07-03

**Soundness:** 2 fair
**Presentation:** 3 good
**Contribution:** 2 fair
**Rating:** 5
**Confidence:** 4

**Summary:**

Causal effect estimation from data often requires assumptions about the causal relationships, either through explicit causal graph structures or implicit conditional independence statements. When confounding exists, the front-door adjustment becomes important for estimating the causal effect of treatment on the outcome using post-treatment variables. This paper studies testable conditional independence statements to compute causal effects using a front-door-like adjustment without knowing the graph under limited structural information. The effectiveness of the method is demonstrated through experiments on both random graphs and real-world causal fairness benchmarks.

**Strengths:**

1. The proposed method enables estimating causal effects without requiring knowledge of the causal graph. Instead, it utilizes front-door-like adjustments based on post-treatment variables, making it applicable even in scenarios with unobserved confounding.
2. The proposed method relies on conditional independence statements that can be directly tested from observational data. This allows for identifying causal effects using observable information without the need for specifying the entire causal graph.
3. The proposed method requires only limited structural side information, which can be obtained from an expert. This requirement is less demanding than specifying the entire causal graph, making the approach more practical and feasible.



**Weaknesses:**

1. The algorithm presented in the paper relies on stronger assumptions, but the paper does not mention them, raising doubts about the soundness and completeness of the proposed method. My main doubts are listed in the Questions.

2. Figure 5 appears to have a mislabeled Y-axis. It seems to be "Average ATE errors".


Minors:
Line 59, criteria -> criterion



**Questions:**

1. Assumption 1 alone may not be sufficient. It might also be necessary to explicitly state that variable y is not a child node of variable t.

2. TTheorem 3.1 appears to be incomplete. For example, we have $t\rightarrow b\rightarrow y$ and $t\leftrightarrow y$, where $\leftrightarrow$ denotes a latent confounder. In this case, b is a cause of y, and the conditional independence between b and y may no longer hold.

3. Some cases present challenges in identifying P(Y|do(t=t)) due to the complexity introduced by latent confounders. It is unclear how to exclude these non-identification cases and handle them appropriately.

---

> ### Author Rebuttal · Authors · 2023-08-09
>
> We thank the reviewer for their valuable feedback and time. Below, we respond to their comments:
>
> 1. **Figure 5**:
>
> We thank the reviewer for pointing out this typo. We will correct this in the revised version.
>
> 2. **It might be necessary to explicitly state that $y$ is not a child node of $t$**:
>
> We thank the reviewer for bringing this up. First, note that if $y$ is a child of $t$ and a bi-directed edge exists between $t$ and $y$ (as stated in line 160), the causal query is not identifiable (see Theorem 4 of Tian and Pearl 2002, *A General Identification Condition for Causal Effects*). Moreover, in this scenario, our desired conditional independencies will not pass as there does not exist a set $\mathbf{z}$ for which $b \perp y | \mathbf{z}, t$ since $y \in b$. Thus, our algorithm will correctly return “I don’t know” as the answer. We will add a remark in the revised version to make this explicit.
>
> 3. **Theorem 3.1 appears to be incomplete ($b \perp y | \mathbf{z}, t$ may not hold for the graph $t \leftrightarrow  y; t \rightarrow  b \rightarrow  y$)**:
>
> We thank the reviewer for mentioning this case. We note that for the traditional front-door graph that the reviewer suggested, our available side information only reveals the edges $t \leftrightarrow y$ and $t \rightarrow b$, i.e., it does not reveal the edge $b \rightarrow y$ as we do not assume the knowledge of the entire graph. **This is not sufficient to identify the causal effect from observational data as the edge between $b$ and $y$ cannot be oriented.** Specifically, the traditional front door graph is not distinguishable from the following two graphs: (a) $t \leftrightarrow  y; t \rightarrow  b \leftarrow  y$  and (b) $t \leftrightarrow  y; t \rightarrow  b \leftrightarrow  y$. For these two graphs, we have $\mathbb{P}(y|do(t)) = \mathbb{P}(y)$ which is different from the front-door adjustment formula. Therefore, yes, our algorithm cannot identify the causal effect in the traditional front-door graph but correctly so, since it is not identifiable from the available side information. We will include this discussion in the revised version.
>
> 4. **Complexity introduced by latent confounders**:
>
> We note that our conditions are sufficient irrespective of the number or complexity of the latent confounders. This means that if a causal query is not identifiable, then our conditions will not hold. Thus, the algorithm may say “I don’t know” but it will not make a mistake. For more examples of graphs with multiple latent confounders beyond Figure 2, please have a look at Figure 8 and Figure 11 in the supplementary.
>
> ---
> We hope that our response addresses the reviewer's concerns and that they would consider increasing their score.

---

> > ### Comment · Reviewer_XvUT · 2023-08-17
> > **Response**
> >
> > Thanks for your response and clarification. I have read the authors' rebuttal and other reviewers' comments.
> > I will maintain my rating.

---

> > > ### Author Response · Authors · 2023-08-21
> > >
> > > We thank the reviewer for reading our rebuttal as well the comments of other reviewers. We are glad that the reviewer's questions were clarified.

---

### Official Review · Reviewer_Mu2q · 2023-07-05

**Soundness:** 4 excellent
**Presentation:** 3 good
**Contribution:** 3 good
**Rating:** 7
**Confidence:** 4

**Summary:**

The authors proposed a method for estimating causal effects without requiring the knowledge of fully-specified causal graph, focusing on the case where unobserved confounding between treatment and outcome exists. This approach using a front-door-like adjustment formula has a novel contribution in that it can estimate causal effect using only simple structural side information which can be obtained from an expert and is less demanding than specifying the entire causal graph. The authors provide sufficiency proofs and demonstrate clear graphical criteria (a generalized front-door condition) for the proposed front-door-like adjustment formula.


**Strengths:**

The authors present a generalized formula that accounts for the variability of the front-door criterion based on the structure of the graph. They provide sufficient conditions clearly for the formula and demonstrate its validity. Hence, in realistic scenarios where unobserved variables may exist between treatment and outcome, this methodology can be effectively utilized, proving its utility. In order to facilitate understanding for readers, the paper includes comprehensive prerequisite knowledge.
it is anticipated that the formula proposed in this paper will have high utility.

**Weaknesses:**

No specific weakness.

**Questions:**

I once worked on the same problem a few years ago when I first read Entner’s paper on backdoor criterion but I had no luck. So I am super happy to read this paper!


I am wondering whether you can compute the variance of ATE estimation for each selection of S so that we can make use of inverse variance weighting instead of simple average. Further, can you employ double machine learning like approach? (e.g., Jung et al. 2021, https://ojs.aaai.org/index.php/AAAI/article/view/17438)


Suggestion:
In consideration of the importance of Theorem 3.2 as a sufficient condition, it is recommended to include brief proof sketch not only in the appendix but also in the main body of the paper.



**Limitations:**

The author clearly presents the limitations of the methodology proposed in Appendix A.2. As stated by the author, assumption 3 among the three assumptions is undoubtedly a strong assumption, which would require substantial domain knowledge to satisfy it in practical situations. Therefore, it is necessary to exercise caution and ensure sufficient attention when applying the methodology suggested by the author in experimental settings.

---

> ### Author Rebuttal · Authors · 2023-08-09
>
> We thank the reviewer for their valuable feedback and time. Below, we respond to their comments:
>
> 1. **Using inverse variance weighting + double machine learning**:
>
> We thank the reviewer for pointing out the possibility of variance and bias reduction using inverse variance weighting and double machine learning. Although it might be hard to non-parametrically estimate the variance for each selection S, perhaps it is possible with parametric assumptions such as linearity. We will add this remark as a possibility in the camera-ready version.
>
> 2. **Proof sketch of Theorem 3.2**:
>
> We thank the reviewer for their suggestion and will include a proof sketch of Theorem 3.2 in the main body in the revised version.

---

> > ### Comment · Reviewer_Mu2q · 2023-08-12
> >
> > Thank you for your response. Other reviewers comments and the authors' responses are satisfactory. I will maintain my score.

---

> > > ### Author Response · Authors · 2023-08-21
> > >
> > > We thank the reviewer for reading our rebuttal as well the comments of other reviewers. We are glad that the reviewer found our response satisfactory.

---

### Official Review · Reviewer_amuY · 2023-07-12

**Soundness:** 4 excellent
**Presentation:** 4 excellent
**Contribution:** 3 good
**Rating:** 5
**Confidence:** 4

**Summary:**

The paper investigates the problem of estimating the average treatment effect of variable "t" on variable "y" within the Pearlian framework. The paper proposes an algorithm that enables causal effect estimation using a front-door-like criterion while relying on only a limited knowledge about the underlying graph structure.

The core of the algorithm lies in the search for a subset of obserables "z" that satisfies a series of independence criteria, thereby establishing a front-door-like formula using "z". The proofs employed in the paper leverage the do-calculus and the identifiability criterion of Tian and Pearl. In addition to its theoretical contributions, the paper also presents empirical demonstrations of the proposed approach across three distinct categories: (1) random Structural Causal Models; (2) synthetic data; and (3) real-world fairness benchmarks.

**Strengths:**

The paper addresses an important problem in the field of causality research by examining the limitations of existing algorithms for causal effect estimation. It specifically aims at improving on the assumption of having access to the underlying causal graph, which is often not readily available. The main contribution of the paper is the introduction of a method for identifying a set of observables 'z' that enables the generation of a front-door-like formula, thereby improving causal effect estimation under limited graph availability.

The paper presents a clear problem statement and provides well-presented proofs. By offering an alternative perspective on causal effect estimation, the paper provides valuable insights for tackling this challenging problem.  Overall, the paper makes a meaningful contribution to the field and opens avenues for further research.

**Weaknesses:**

One potential weakness of the paper is that once the requirement of designing a subset "z" satisfying the front-door-like criterion (Eqn (9)) is fixed, the proofs and the proposed independence criteria are relatively straightforward and achievable using the rules of do-calculus and the identifiability criterion.

It is also not convincing to me how and why the exhaustive search runs fast for the random class of graphs generated in Section 4.1.  The expected number of unobservables is of the order O(p), I am curious to know why the bidirected edges are chosen with probablity q/p?  It would be convincing to see positive results for larger p with more unobservables.

The in-completeness of the proposed algorithm and the mandatory requirement of Assumption 2 are the two major drawbacks of the paper (Please refer to Limitations for a detailed discussion.)

**Questions:**

As pointed out by the authors, Assumption 2 is mandatory for the approach to work. This observation is not surprising but somewhat disheartening. It would be valuable to explore if there are any workarounds or alternative approaches to testing this assumption, possibly utilizing Confidence Interval (CI) tests. Given that the available knowledge is limited to only the children of "t," I am not sure if this is possible to test. Further exploration or discussion on potential alternatives or extensions to address this limitation would enhance the paper's robustness.

Regarding the choice of the random class of graphs for the benchmark, it would be beneficial to have an explanation or justification provided in the paper. Understanding the rationale behind the class of graphs considered would provide more insights.

I also suggest to provide a more detailed description of Algorithm 1 in the main paper, as that would enable readers to have a better understanding of the algorithm.

**Limitations:**

The primary limitation of the paper, in my opinion, is regarding the completeness of the algorithm. The fact that the proposed algorithm is not complete represents a significant drawback. A complete algorithm would have provided a more robust and comprehensive solution to the problem.

As explained in the paper, Assumption 2 is crucial in order for the search to work which is another crucial downside of this approach.  This reliance on a critical assumption may limit the generalizability of the approach to real-world scenarios where such assumptions may not hold.  Indeed, I feel like finding a workaround or alternative approach to mitigate the reliance on Assumption 2 would greatly enhance the paper's technical as well as practical value.

I believe issues regarding sample complexity are out of the scope of the paper and perhaps be considered for future work.

---

> ### Author Rebuttal · Authors · 2023-08-09
>
> We thank the reviewer for their valuable feedback and time. Below, we respond to their comments:
>
> 1. **How and why does the exhaustive search run fast for the random class of graphs generated in Section 4.1?**:
>
> We thank the reviewer for this question. We do not claim that the exhaustive search runs fast for the class of random graphs considered in Section 4.1 and we will clarify this in the revised version. The worst-case run-time would still be exponential in $p$.
>
> 2. **Why are the bi-directed edges chosen with probability $q/p$?**:
>
> We choose $q/p$ to be the probability with which a bi-directed edge exists between two observed nodes so that the expected number of unobserved nodes is $q/p \times p(p-1)/2 = q(p-1)/2$ as the reviewer mentioned. As per the reviewer’s suggestion, we increased the number of unobserved nodes by choosing $q$ to be the probability with which a bi-directed edge exists between two observed nodes. In this case, the expected number of unobserved nodes is $qp(p-1)/2$. We ran simulations with $q \in$ {0.1, 0.15, 0.2}. As expected, with more confounding, there are almost no conditional independence (CI) tests to exploit, yet there are (roughly 1% in all 3 settings) graphs where our approach still applies. We note that any causal discovery-based approach is expected to suffer even more without any meaningful collection of CIs. We will add these results in the revised version.
>
> 3. **Rationale behind the class of graphs considered**:
>
> We thank the reviewer for the suggestion to add more explanation regarding the choice of random class of graphs. Our choice ensures that every node has the same bounded in-degree in expectation. This is common in recent works, e.g., Addanki et al. 2020, *Efficient Intervention Design for Causal Discovery with Latents*, where every directed edge is added with the same probability as in our work. We are happy to address any more questions that the reviewer may have.
>
> 4. **Detailed description of Algorithm 1**:
>
> We thank the reviewer for their suggestion and will include a detailed description of Algorithm 1 in the main body in the revised version. The variable $c_1$ is used to perform an average over different train-test splits. The variables $c_2$, $ATE^r_z$, and $ATE^r_s$ are used to perform an average over different subsets $\mathbf{z}$ that satisfy our conditions for a specific train-test split. We will also add more explanation on how the two equations in the algorithm are equivalent to the first moment versions of (9) and (10), respectively.
>
> 5. **Completeness of the algorithm**:
>
> To show the completeness of Theorem 3.2, one would have to show the following -- for every class of graphs where our structural side information holds but our algorithm fails, (i.e., at least one the conditional independencies in (7) or (8) fail to hold), the causal effect is not unique, i.e., the causal effect cannot be identified by the front-door-like formulae in (9) and (10) while only using observational data and not knowing the underlying causal graph.
>
> Typically, this is done by explicitly constructing two causal graphs where the algorithm fails and the causal effects have different values. For example, Shpitser et al. 2008, Complete Identification Methods for the Causal Hierarchy, showed the completeness of their ID algorithm this way through explicit constructions. It is highly non-trivial to show the completeness of our algorithm using a similar approach where there are no specific graphs at hand. However, we do show the importance of (8) via the causal graphs in Figure 3. These graphs are such that our structural side information holds, (7) holds for both, but (8) only holds for the bottom one. Then, we show that causal effects formulae for these graphs are indeed different (see lines 195-215). While we agree that providing additional examples towards full completeness of our algorithm would strengthen our results, we reserve this for future work as it is a highly non-trivial extension.
>
> 6. **Workaround for Assumption 2**:
>
> We thank the reviewer for their comment about Assumption 2. In practice, we believe this assumption is more likely to be true than not. But, we agree with the reviewer that this assumption is a limitation and working around this assumption is an interesting future work. We believe that our results could be derived under the weaker condition that there is a back-door path between $t$ and $y$ which is not blockable. On the other hand, if there is no unblockable back-door path between $t$ and $y$, it may be easier to find back-door adjustment sets. We will append this to the discussion on Assumption 2 in the limitations section (Appendix A.2).
>
> ---
> We hope that our response addresses the reviewer's concerns and that they would consider increasing their score. We believe it is indeed an advantage that our proofs are straightforward and clear in hindsight. However, note that it is not clear a priori that the proposed independence criteria would lead to a single (front-door-like) causal effect formula that spans multiple causal graphs which are not Markov equivalent as we show in this work.

---

> > ### Comment · Reviewer_amuY · 2023-08-15
> >
> > Thank you for your response.  I would like to maintain my current score.

---

### Official Review · Reviewer_ZMBw · 2023-07-19

**Soundness:** 3 good
**Presentation:** 3 good
**Contribution:** 2 fair
**Rating:** 5
**Confidence:** 2

**Summary:**

This paper proposes a method for estimating causal effects between the treatment variable and the outcome variable using front-door adjustment beyond the Markov equivalence class. This method is applicable when there are unobserved confounders between the treatment and outcome variables and does not require knowledge of the entire causal graph, but only limited graph knowledge. The authors introduce three assumptions and the causal identifiability theorem and the generalized front-door condition to achieve the estimation of the Average Treatment Effect (ATE). Through experiments, the paper demonstrates that the proposed framework provides identifiability in random fusion compared to PAG-based algorithms, exhibits lower error rates in ATE estimation compared to baseline, and shows practical applicability in causal fairness analysis.

**Strengths:**

- The authors propose testable conditional independence statements for front-door-like adjustment without graph knowledge under limited structural side information.
- The experimental results show that the proposed method is effective on random graphs and real causal fairness benchmarks.

**Weaknesses:**

- It seems that the identification of the proposed method highly depends on Assumption 3. Assumption 3 requires knowledge of all direct descendant nodes $b$ of the treatment variables, which is too strong and difficult to achieve in practical scenarios.
- Compared to PAG-based algorithms, the proposed method in this paper proves its ability to effectively provide identifiability. However, it requires expert knowledge to provide structural information, which may not necessarily demonstrate better applicability than PAG-based methods.

**Questions:**

See above.

---

> ### Author Rebuttal · Authors · 2023-08-09
>
> We thank the reviewer for their valuable feedback and time. Below, we respond to their comments:
>
> 1. **Assumption 3 requires knowledge of all direct descendant nodes**:
>
> We agree with the reviewer that requiring the knowledge of all the children of the treatment variable is crucial for our method. An important future direction could be to alleviate this requirement, say, to knowing only a subset of the children. For example, one could think of approximating the causal effect when only the children corresponding to weak edges are unknown. Such variations around our condition are promising directions for future work. We will append this to the discussion on Assumption 3 in the limitations section (Appendix A.2).
>
> 2. **Comparing the applicability of our method to PAG-based methods**:
>
> We thank the reviewer for bringing up this great point. We agree that in scenarios where the structural side information is not available, we may have to resort to PAG-based methods. We will clarify this in the revised version. Additionally, we note that the FCI algorithm (that produces a PAG) involves a sequence of adaptive conditional independence (CI) tests where the choice of the next test depends on the previous ones. Specifically, orientation stages tend to propagate errors in non-trivial ways (for example, https://arxiv.org/pdf/1607.03975.pdf shows how to handle these for the PC algorithm which has only three orientation rules). This gets very involved for the FCI algorithm which has many orientation rules and makes it difficult to control the false discovery rate for PAG based methods. In contrast, the CI tests involved in our method could be carried out in parallel and therefore require little-to-no adaptivity. Thus, our method can be viewed as a way to mitigate the issues associated with adaptive testing by using structural side information.
>
> ---
> We hope that our response addresses the reviewer's concerns and that they would consider increasing their score.

---

### Decision · Program_Chairs · 2023-09-21

**Decision:**

Accept (poster)

**Comment:**

This paper shows that one can compute causal effects using front-door adjustment even when the underlying causal graph structure is not fully known. The reviewers were excited by the progress made by this work, and I think this paper will open the door to considering more general situations where limited side-information is available.